# SimuAgent: An LLM-Based Simulink Modeling Assistant Enhanced with Reinforcement Learning

## Abstract

Large language models (LLMs) have revolutionized text-based code automation, but their potential in graph-oriented engineering workflows remains underexplored. We introduce *SimuAgent*, an LLM-powered modeling and simulation agent tailored for Simulink. SimuAgent replaces verbose XML with a concise, dictionary-style Python representation, dramatically cutting token counts, improving interpretability, and enabling fast, in-process simulation. A lightweight plan–execute architecture, trained in two stages, equips the agent with both low-level tool skills and high-level design reasoning. To tackle sparse rewards in long-horizon tasks, we propose *Reflection-GRPO* (ReGRPO), which augments Group Relative Policy Optimization (GRPO) with self-reflection traces that supply rich intermediate feedback, accelerating convergence and boosting robustness. Experiments on *SimuBench*, our newly released benchmark comprising 5300 multi-domain modeling tasks, show that a Qwen2.5-7B model fine-tuned with SimuAgent converges faster and achieves higher modeling accuracy than standard RL baselines, and even surpasses GPT-4o when evaluated with few-shot prompting on the same benchmark. Ablations confirm that the two-stage curriculum and abstract-reconstruct data augmentation further enhance generalization. SimuAgent trains and runs entirely on-premise with modest hardware, delivering a privacy-preserving, cost-effective solution for industrial model-driven engineering. SimuAgent bridges the gap between LLMs and graphical modeling environments, offering a practical solution for AI-assisted engineering design in industrial settings.

## 1 Introduction

Simulink has become the de facto standard for model-based design in safety-critical industries, with over five million engineers relying on it for developing automotive, aerospace, and energy systems (MathWorks, 2023). Major industry players from Tesla to Boeing have deeply integrated Simulink into their workflows, while stringent safety standards like ISO 26262 and DO-178C explicitly recommend or mandate its use for compliance (ISO, 2018; RTCA, 2011). This entrenched position means that even modest improvements in modeling efficiency can yield substantial economic benefits and accelerate certification processes—addressing the urgent need to reduce the high costs of developing complex models.

Beyond its industrial dominance, Simulink presents unique technical challenges for large language models (LLMs). Unlike text-based programming, Simulink employs a hierarchical, graphical paradigm with complex block diagrams, signal routing, and strict topological constraints. This makes it an ideal testbed for evaluating and advancing LLM reasoning capabilities in highly structured, non-textual domains. Moreover, Simulink serves as a crucial bridge to address a fundamental limitation of current LLMs: their disconnection from the physical world. Trained almost exclusively on text, LLMs lack grounded understanding of physical laws, causality, and dynamic processes (Wang et al., 2023b). Simulink provides a "physics sandbox" where LLMs can model, simulate, and interact with systems spanning control logic, mechanics, electronics, and thermodynamics—enabling them to move beyond abstract textual knowledge toward embodied understanding of how physical principles govern real-world systems.

Developing LLMs for graphical modeling environments faces three core challenges. First, **data scarcity**: publicly available Simulink model datasets are extremely limited (Zhang et al., 2025), making direct training of large models challenging. Second, **syntactic and semantic constraints**: graphical models must adhere to strict structural rules (e.g., connection validity, parameter consistency), and LLMs can easily violate these constraints or hallucinate non-existent blocks (Shrestha & Csallner, 2021). Third, **context window limitations**: the descriptions of complex models (e.g., industrial systems with hundreds of blocks) can far exceed the LLM's context window, necessitating strategies like modular generation or hierarchical refinement.

Recent exploratory work has begun to investigate LLM applications in Simulink environments. SLGPT fine-tunes GPT-2 to translate text into Simulink's XML format, inadvertently uncovering toolchain defects (Shrestha & Csallner, 2021). For model analysis, a requirements-driven slicing technique was proposed to textualize Simulink models for LLM processing (Luitel et al., 2024). In mutation testing, BERTiMuS adapts CodeBERT to generate model variants for requirements-aware testing (Zhang et al., 2025). While these pioneering efforts demonstrate the feasibility of applying LLMs to Simulink-related tasks, they represent only isolated solutions and remain at a prototypical stage rather than a comprehensive modeling assistant. Developing an LLM-driven agent capable of interacting with engineers, understanding context, and providing proactive support throughout the entire modeling workflow remains a significant gap in current research.

Although LLMs excel in reasoning and coding (Guo et al., 2024; 2025), they often suffer from hallucinations (Zhang et al., 2023) in scenarios requiring specialized knowledge, such as the domain-knowledge-intensive field of Simulink. To mitigate this, a common approach is to enable LLMs to retrieve external information via tool invocation (Schick et al., 2023), typically through prompting strategies (e.g., IRCoT (Trivedi et al., 2022), ReAct (Yao et al., 2023)) or fine-tuning (e.g., Toolformer (Schick et al., 2023)). However, these methods often depend on high-quality, manually annotated trajectories that are difficult to obtain at scale. Reinforcement Learning (RL) (Kaelbling et al., 1996; Sutton et al., 1999) offers another viable path for enhancing LLM tool use and reasoning capabilities; recent studies show RL can enable LLMs to learn complex reasoning skills solely from environmental rewards (Guo et al., 2025). To simplify RL tuning, direct optimization methods (Rafailov et al., 2023; Meng et al., 2024) and alternatives like Group Relative Policy Optimization (GRPO) (Shao et al., 2024) have emerged, the latter estimating baselines from group scores, obviating the need for a critic model. However, RL still faces sparse rewards in tool interaction—models receive feedback only upon final success despite requiring multiple reasoning steps. Inspired by Reflexion (Shinn et al., 2023), which achieves improvements through linguistic feedback but lacks any parameter update mechanism, we integrate reflection directly into GRPO training: our approach generates reflection traces from environmental feedback and tool results, enabling the model to learn from richer signals during parameter optimization, thus accelerating convergence and improving reasoning accuracy.

**Our Contributions.** To advance LLM-driven graphical modeling, we propose *SimuAgent*—a plan-execute agent framework operable on laptop-grade GPUs, designed to drive the full Simulink modeling, analysis, and fine-tuning loop end-to-end. Its key contributions are:

1. **SimuAgent Framework and Lightweight Representation:** SimuAgent transforms Simulink models into a compact Python dictionary (JSON) format that LLMs can process more efficiently. This lightweight representation compresses verbose XML while preserving hierarchical model structure, significantly reducing token consumption and improving interpretability. Integrated with an in-process Python test harness, SimuAgent enables instant structural validation and parameter tuning. For complex designs, it automatically encapsulates related modules into subsystems to simplify modeling.

2. **Reflection-GRPO (ReGRPO):** We introduce ReGRPO, an enhanced version of GRPO, which incorporates *reflection traces*—automatically generated feedback derived from discrepancies with reference models and tool invocation results. This provides rich training signals beyond binary success, accelerating convergence and improving robustness in sparse-reward settings.

3. **Abstract–Reconstruct Data Augmentation:** To strengthen model abstraction and generalization, we propose a self-supervised Abstract–Reconstruct augmentation strategy. The agent first generates a structured summary from a Simulink model, then attempts to reconstruct the original model based solely on that summary. This loop teaches the agent to map

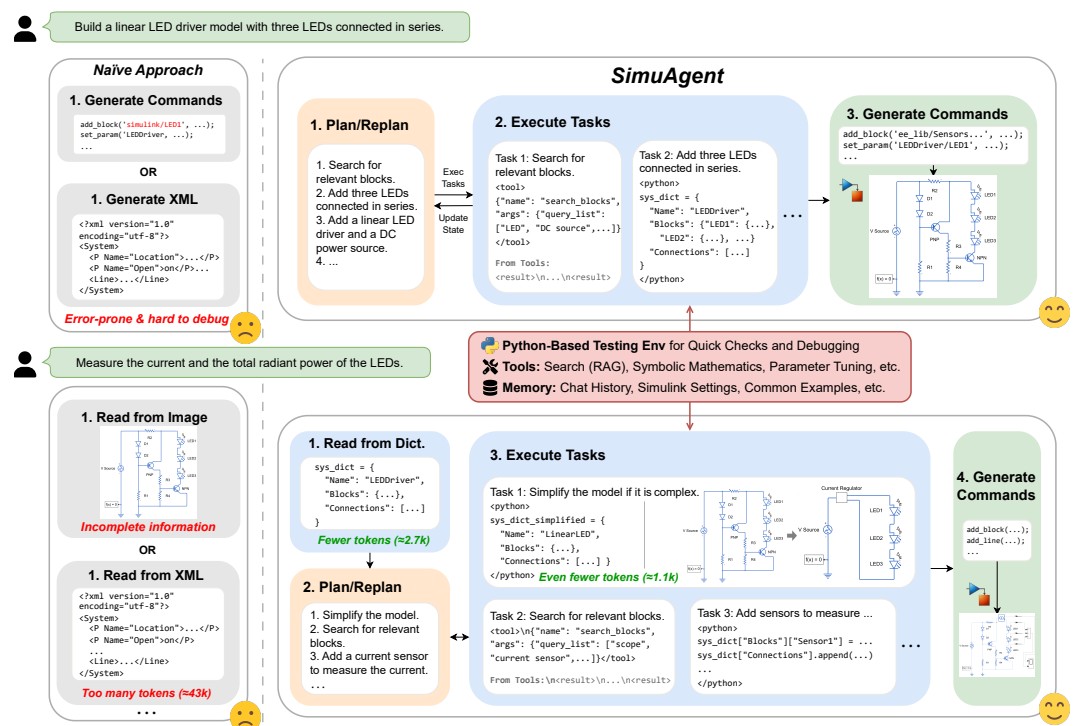

Figure 1: Comparison between SimuAgent and conventional workflows.

between high-level descriptions and low-level implementations, enhancing its capacity to reason about and synthesize complex system architectures.

4. **SimuBench Public Benchmark:** We release *SimuBench*, the first large-scale benchmark for LLM-based Simulink modeling. It comprises 5300 tasks across control, mechanical, electrical, fluid, thermal, and electromagnetic domains. Tasks include model creation, modification, and question-answering (Q&A), with complete source files, XML representations, and visualizations for reproducible evaluation.

## 2 SIMUAGENT ARCHITECTURE

*SimuAgent* is an LLM-powered agent framework for the modeling and simulation of multi-domain physical systems. It targets the automation of model construction, analysis, and validation by combining **lightweight representation**, a **plan-execute** paradigm, and **rapid verification**, as sketched in Figure 1.

**Overall architecture.** SimuAgent adopts a streamlined plan–and–execute design (Wang et al., 2023a; Shen et al., 2023). Unlike complex multi-agent LLM systems (MAS), this approach avoids burdensome role assignment, keeps the prompt context compact, and is far easier to fine-tune for diverse objectives. Recent studies further indicate that MAS configurations do not consistently deliver gains on standard benchmarks (Pan et al., 2025), reinforcing our choice of a lean architecture.

**Python-based model representation and validation environment.** SimuAgent introduces a JSON-compatible **Python dictionary** for representing Simulink models (blue region in Figure 1). Compared with screenshots or XML files, the dictionary offers three clear benefits:

- **Compact, semantically focused.** Only essential details—block names, key parameters, and connectivity—are retained, while visual coordinates and styling are discarded. This slashes token counts: in Figure 1, the XML uses $\sim$ 43k tokens, whereas the dictionary needs just $\sim$ 2.7k.
- **Naturally LLM-friendly.** Because LLMs handle Python effortlessly, they can generate or edit the dictionary directly, with no format conversion or elaborate prompting.

- **Fast simulation and debugging.** Models encoded as dictionaries can be validated inside pure Python, eliminating repeated MATLAB Engine calls and their attendant overhead. After passing validation, they are ported to Simulink for high-fidelity runs.

**Simplification and hierarchical structuring of complex systems.** For large, tightly coupled models, SimuAgent automatically clusters internally related blocks that are irrelevant to the current task, encapsulating them into higher-level modules with concise summaries—akin to `Subsystems` or `Area Annotations` in Simulink. Parameters unrelated to the task may be omitted. This purely representational simplification boosts readability, reduces both visual and token complexity, and lowers LLM inference cost, yet leaves the underlying Simulink hierarchy unchanged.

**Lightweight Python testing environment.** To curb the expense of frequent MATLAB Engine calls, SimuAgent includes a local Python-based testing environment testbed that performs static checks on signal types, parameter ranges, and port wiring. Detected issues trigger immediate warnings, allowing rapid correction before invoking MATLAB for full simulation. The same testbed supports batch edits and fast compile–feedback loops, keeping interaction snappy during model development.

**Integration with external tools.** SimuAgent ties together multiple toolchains for an extensible, highly automated workflow. A RAG-based knowledge base enables semantic search over common Simulink blocks; symbolic-math utilities translate equations into executable blocks; and parameter-tuning modules optimise controllers for performance and robustness. When only a final answer is needed, an entire reasoning sub-task can be wrapped as a standalone tool call, sparing the main dialogue from unnecessary intermediate steps.

## 3 SimuAgent Training Framework

To boost both the performance and generalisation capability of SimuAgent, we design an enhanced training framework that tackles data sparsity and inefficient exploration in complex system design. The framework integrates three key ideas: 1) **Staged training** that matures the agent's skills from execution to high-level planning; 2) **Abstract–Reconstruct data augmentation** that converts abundant raw Simulink models into self-supervised training pairs; and 3) **ReGRPO** that supplies richer learning signals than scalar rewards.

### 3.1 Staged Training Strategy

We adopt a two-stage curriculum that progressively strengthens the same language model inside the Simulink environment.

**Stage 1: Execution focus.** The agent learns to invoke tools and complete low-complexity tasks such as creating small systems, editing modules, and tuning parameters. These actions require limited reasoning and provide dense feedback, making them ideal for bootstrapping.

**Stage 2: Planning integration.** The curriculum then shifts to larger models and prompts that *encourage*—but do not force—the agent to plan, decompose, and simplify tasks when beneficial. Here the model must develop higher-order skills: (re)planning, architectural design, abstraction, and modular construction. All tokens generated in the main dialogue are used for gradient updates; however, LLM calls executed *outside* that context (e.g. as isolated tool invocations) are excluded because credit assignment is ambiguous and their singleton nature precludes the grouping needed by GRPO/ReGRPO.

### 3.2 Abstract–Reconstruct Data Augmentation

Structured summaries and documentation for Simulink projects are scarce, yet such abstractions are crucial for teaching a model to reason about complex systems. We therefore propose a self-supervised **Abstract–Reconstruct** loop, inspired by VAE-style reconstruction.

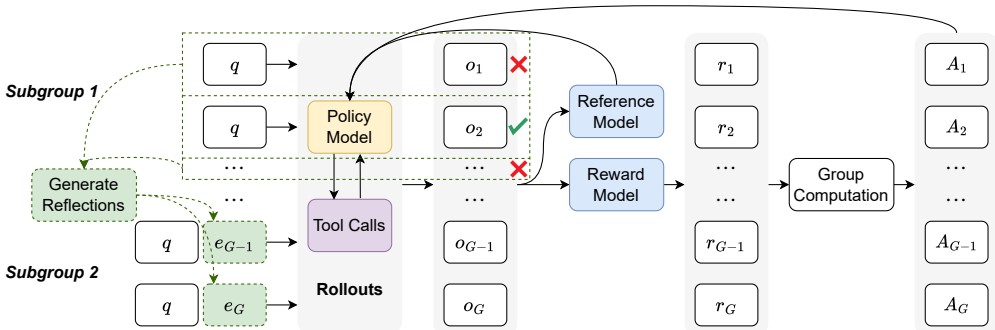

Figure 2: ReGRPO architecture. Black solid arrows show the original GRPO pipeline. The green dashed lines and boxes highlight ReGRPO's extensions: the group is split into two subgroups—after Subgroup 1 completes rollouts, reflections on failed rollouts are generated and shared with Subgroup 2 to guide its policy.

1. **Abstract generation.** Given a Simulink model, SimuAgent produces a structured summary or textual documentation, training its ability to distil key architecture and behaviour.

2. **Model reconstruction.** Conditioned *only* on that summary, the agent attempts to rebuild the original model, reinforcing its capacity to synthesise concrete designs from abstract specifications.

The loop is evaluated along two axes: *(i) Structural completeness and fidelity*—does the reconstruction preserve essential modules and connections, measured by structural comparison? *(ii) Functional correctness*—does the reconstructed model execute successfully in simulation, indicating behavioural consistency?

Abstract-Reconstruct is introduced in Stage 2, after the agent has already mastered basic construction tasks in Stage 1, ensuring stable learning during this more demanding phase.

### 3.3 REFLECTION-GRPO (REGRPO)

**Group Relative Policy Optimization (GRPO).** To improve the stability of policy optimization and remove the need for an explicit value function, Shao et al. (2024) introduce GRPO. For each input question $q$, the reference policy $\pi_{\text{ref}}$ samples a group of responses $\{o_1, \ldots, o_G\}$. The policy model $\pi_\theta$ is then optimized by maximizing

$$\mathcal{J}_{\text{GRPO}}(\theta) = \mathbb{E}_{q \sim \mathcal{D}, \{o_i\} \sim \pi_{\text{old}}(\cdot|q)} \frac{1}{G} \sum_{i=1}^{G} \frac{1}{|o_i|} \sum_{1 \leq t \leq |o_i|} \left\{ \min \left[ \frac{\pi_\theta(o_{i,t}|q, o_{i,<t})}{\pi_{\text{old}}(o_{i,t}|q, o_{i,<t})} \hat{A}_{i,t}, \right. \right.$$

$$\left. \left. \text{clip} \left( \frac{\pi_\theta(o_{i,t}|q, o_{i,<t})}{\pi_{\text{old}}(o_{i,t}|q, o_{i,<t})}, 1 - \epsilon, 1 + \epsilon \right) \hat{A}_{i,t} \right] - \beta \, \mathbb{D}_{\text{KL}} \left[ \pi_\theta || \pi_{\text{ref}} \right] \right\}, \quad (1)$$

The advantage $\hat{A}_{i,t}$ is computed *within* a group, so the average group reward acts as a baseline and no extra value network is required. Hyper-parameters $\epsilon$ and $\beta$ control the clipping range and KL regularisation strength, respectively.

**Reflection-GRPO (ReGRPO).** GRPO can struggle on sparse-reward tasks because a single scalar reward offers little guidance. ReGRPO augments GRPO with a **Reflection Mechanism** that supplies rich textual feedback:

1. **Two-subgroup architecture.** For each input, the first subgroup attempts the task with the normal prompt. If its output falls short, the model reflects on its reasoning trace, compares it with the reference answer (or environment feedback), and distils the key insights as *reflection text*.

2. **Guided exploration.** The reflection text $e_i$ is concatenated with the original input and passed to a second subgroup, which re-attempts the task with more focused exploration. Keyword filtering and other safeguards prevent leakage of ground-truth answers, and

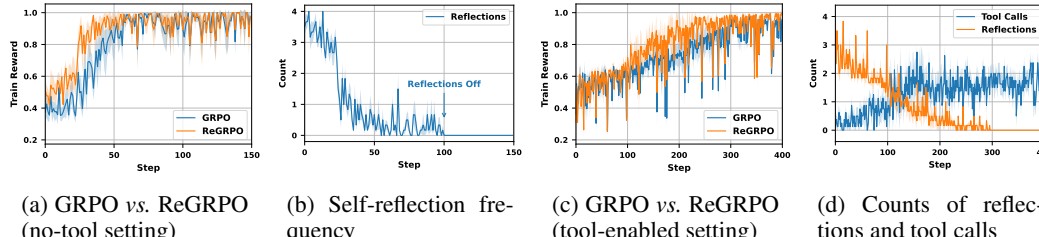

(a) GRPO *vs.* ReGRPO (no-tool setting)

(b) Self-reflection frequency

(c) GRPO *vs.* ReGRPO (tool-enabled setting)

(d) Counts of reflections and tool calls

Figure 3: Comprehensive ablation study of ReGRPO training dynamics on the GSM8K dataset. (a) With tool calls disabled, ReGRPO achieves higher returns earlier than vanilla GRPO but shows mild late-stage oscillations; GRPO learns more slowly yet remains stable. (b) The number of self-reflections per episode rapidly diminishes as the agent gains task proficiency, signalling reduced deliberation overhead. (c) When tool invocations are permitted, ReGRPO again exhibits superior sample-efficiency, though both methods eventually plateau at comparable performance levels. (d) Over the course of training, the agent gradually substitutes costly self-reflection with a moderate, steady rate of tool usage, converging to a lean and effective decision-making routine. The shaded error bands represent the variance across six runs with different random seeds.

stronger language models can be employed to improve the quality and generality of the reflections.

3. **Adaptive frequency.** Reflection is invoked often during early training for faster progress, then gradually decays—e.g. via non-replacement sampling or a probabilistic schedule—so that mature policies learn to solve tasks autonomously. Once performance stabilises, reflection can be disabled entirely, reverting to vanilla GRPO for inference-time compatibility.

**GRPO / ReGRPO with Tool Calls.** Unlike vanilla GRPO, when tool calls are allowed, $o$ now comprises both LLM-generated tokens and the tokens returned by external tools $\mathcal{T}$. We therefore apply an extra masking operation to tokens returned by tools: $I(o_{i,t}) = 1$ if $o_{i,t}$ is produced by the LLM, and $I(o_{i,t}) = 0$ if it is supplied by a tool. This design avoids treating tool outputs as learning targets, which may lead to unintended learning dynamics, and instead allows the LLM to focus on *how* to use these tools effectively. Recent work on search-augmented LLMs (Jin et al., 2025) also employs a similar masking scheme and reports higher performance than training without masking. The training objective for GRPO/ReGRPO with tool calls becomes

$$
\mathcal{J}_{\text{Tool}}(\theta) = \mathbb{E}_{\substack{q \sim \mathcal{D}, \{o_i\} \sim \pi_{\text{old}}(\cdot|q;\mathcal{T}) \\ \{e_i\} \sim \pi_{\text{old}} \text{ or } e_i = \text{``"}}} \frac{1}{G} \sum_{i=1}^{G} \frac{1}{\sum_{t=1}^{|o_i|} I(o_{i,t})} \sum_{\substack{1 \le t \le |o_i| \\ I(o_{i,t})=1}} \left\{ \min\left[ \frac{\pi_\theta(o_{i,t}|q,e_i,o_{i,<t};\mathcal{T})}{\pi_{\text{old}}(o_{i,t}|q,e_i,o_{i,<t};\mathcal{T})} \hat{A}_{i,t}, \right.\right.
$$

$$
\left.\left. \text{clip}\left( \frac{\pi_\theta(o_{i,t}|q,e_i,o_{i,<t};\mathcal{T})}{\pi_{\text{old}}(o_{i,t}|q,e_i,o_{i,<t};\mathcal{T})}, 1 - \epsilon, 1 + \epsilon \right) \hat{A}_{i,t} \right] - \beta\, \mathbb{D}_{\text{KL}}\left[ \pi_\theta || \pi_{\text{ref}} \right] \right\}, \quad (2)
$$

where $e_i$ represents the reflection text. For GRPO with tool calls, $e_i$ is an empty string. In our ReGRPO framework, the *first* subgroup keeps $e_i$ empty, while the *second* subgroup populates $e_i$ with reflections distilled from a failed rollout sequence generated by the first subgroup. The ReGRPO algorithm is shown in Algorithm 1, where the variable $f$ denotes a prompt template designed to encourage the LLM to reflect. An example of such a template is shown below:

Table 1: Instruction prompt template employed in the reflection stage of ReGRPO training. The placeholder **answer** will be replaced with the reference answer during training.

---

The reference answer is: **answer**, which differs from your previous response. Try to identify which part of your reasoning process or tool usage most likely caused the discrepancy, so you can avoid similar mistakes in the future. Avoid vague statements; instead, give clear and specific insights—for example, noting if a tool was misused, a step was skipped, or an assumption was incorrect. If you truly have no reasonable idea, write exactly: None. Your explanation must be under 150 words and *must not include or refer to the reference answer.*

---

**Algorithm 1** Reflection-GRPO with Tool Calls

**Input** initial policy $\pi_{\theta_{\text{init}}}$; tools $\mathcal{T}$; tasks $\mathcal{D}$; reflection prompt $f$; buffer $\mathcal{E}$; subgroup sizes $G_1, G_2$ ($G_1 + G_2 = G$).

1: $\pi_\theta \leftarrow \pi_{\theta_{\text{init}}}$
2: **for** iteration $= 1, \ldots, I$ **do**
3:     $\pi_{\text{ref}} \leftarrow \pi_\theta, \ p \leftarrow \text{decay}(\text{iteration})$
4:     **for** step $= 1, \ldots, M$ **do**
5:         Sample mini-batch $\mathcal{D}_b \subset \mathcal{D}, \ \pi_{\theta_{\text{old}}} \leftarrow \pi_\theta$
6:         **for all** $q \in \mathcal{D}_b$ **do**
7:             $\mathcal{E} \leftarrow \varnothing$
8:             **for** $i = 1 : G_1$ **(in parallel) do**
9:                 $o_i^{(1)} \sim \pi_{\theta_{\text{old}}}(\cdot \mid q; \mathcal{T})$, obtain reward $r_i^{(1)}$
10:              **if** task failed **then**
11:                 $e \sim \pi_{\theta_{\text{old}}}(\cdot \mid q, o_i^{(1)}, f); \ \mathcal{E} \leftarrow \mathcal{E} \cup \{e\}$
12:             **for** $i = 1 : G_2$ **(in parallel) do**
13:         

$$e_i \leftarrow \begin{cases} \text{sample from } \mathcal{E}, & \mathcal{E} \neq \varnothing \ \wedge \ \text{Uniform}(0,1) < p \\ \text{``''} & \text{otherwise} \end{cases}$$

14:               $o_i^{(2)} \sim \pi_{\theta_{\text{old}}}(\cdot \mid q, e_i; \mathcal{T})$, obtain $r_i^{(2)}$
15:             Compute $\hat{A}_{i,t}$ using rewards $\{r_i^{(1)}\} \cup \{r_i^{(2)}\}$
16:             **for** $k = 1 : \mu$ **do**
17:               Update $\pi_\theta$ by maximizing Eq. equation 2
18:     **Optional:** update reward model if used
19: **return** $\pi_\theta$

(a) Stage 1: GRPO *vs.* ReGRPO

(b) Stage 1: Tool-call and reflection counts

(c) Stage 2: with *vs.* without abstract–reconstruct aug.

(d) Stage 2 from scratch (no Stage 1 pretraining)

Figure 4: Two-stage training analysis on the SimuBench dataset. (a) During Stage 1, ReGRPO learns markedly faster than vanilla GRPO and attains a higher asymptotic reward. (b) Tracing Stage 1 behaviour shows an early surge in both reflections and tool calls; as competence grows, reflections are pruned while a modest, steady tool-invocation rate is retained. (c) In Stage 2, injecting abstract–reconstruct augmented data accelerates convergence and raises the final reward, indicating improved robustness and generalization. (d) Skipping Stage 1 and training Stage 2 from scratch yields noticeably slower learning and a lower plateau, highlighting the benefit of the curriculum.

## 4 EXPERIMENTAL SETUP AND FINDINGS

### 4.1 SIMUBENCH DATASET

To overcome the shortage of standardized benchmarks for LLM-assisted Simulink modeling, we introduce **SimuBench**, a large-scale, multi-domain corpus comprising **5 300 tasks**. Tasks span model creation, editing (adding, deleting, or altering components and parameters), and general model–based QA (e.g. "How many ammeters are present?" "Where is the measurement taken?"). Domains covered include mathematics, control systems, mechanical and electrical engineering, fluid dynamics, thermodynamics, and electromagnetics. Besides official Simulink demos and authoritative open-source engineering textbooks, SimuBench incorporates topologies and parameters from real-world systems, including the Schutterwald natural gas distribution network in Germany (Kisse et al., 2020) and the GB electricity transmission network from the Power Systems Test Case Archive (Bukhsh & McKinnon, 2013).

Table 2: Performance on the SimuBench test dataset (all values in %). Bold indicates best in each column. Small systems $\leq$ 12 blocks; large systems > 12.

| Models | Methods | Small Systems | | | Large Systems | | | |
| --- | --- | --- | --- | --- | --- | --- | --- | --- |
| | | Creat. | Modif. | QA | Creat. | Modif. | QA | Avg. |
| Qwen2.5-7b | Direct Inference | 10.71 | 38.71 | 44.93 | 1.57 | 19.79 | 20.07 | 26.57 |
| | CoT | 9.43 | 27.43 | 42.43 | 1.07 | 20.93 | 20.00 | 23.55 |
| | RAG | 20.71 | 35.50 | 64.43 | 2.93 | 18.71 | 32.93 | 33.70 |
| | SFT | 25.43 | 39.21 | 49.93 | 3.29 | 16.43 | 23.93 | 30.38 |
| GPT-4o | Image | 36.79 | 65.00 | **70.43** | 8.50 | 40.50 | 37.43 | 48.85 |
| (2024-11-20) | XML | 36.79 | **67.29** | 69.29 | 8.50 | 42.29 | 46.21 | 50.45 |
| SimuAgent | No Training | 10.21 | 36.57 | 41.50 | 1.07 | 20.07 | 20.50 | 25.29 |
| | Only Stage 1 | 38.43 | 62.57 | 70.21 | 7.29 | 28.79 | 38.50 | 46.55 |
| | Only Stage 2 | 31.43 | 57.79 | 62.00 | 9.79 | 33.07 | 46.57 | 44.38 |
| | Stage 1 & 2 | **39.71** | 64.93 | 69.07 | **11.71** | **43.43** | **54.50** | **51.89** |

Table 3: Cross-environment generalization performance of ReGRPO (accuracy, %). 'SimuBench' columns report zero-shot transfer from a model trained only on SimuBench. '+FT' denotes minimal domain-specific fine-tuning on that model (607 Modelica / 730 PSCAD examples).

| | General | | Math | | Code | | Modelica | | PSCAD | |
| --- | --- | --- | --- | --- | --- | --- | --- | --- | --- | --- |
| | MMLU | BBH | GSM8K | MATH | HumanEval | MBPP | SimuBench | +FT | SimuBench | +FT |
| **GRPO** | 77.31 | 74.04 | 92.40 | 53.18 | 58.45 | 76.38 | 31.96 | 42.62 | 31.21 | 41.61 |
| **ReGRPO** | 78.14 | 74.73 | 93.30 | 53.91 | 61.89 | 79.83 | 33.84 | 44.54 | 33.04 | 43.49 |

## 4.2 EXPERIMENTAL RESULTS

All experiments employ **Qwen-2.5-7B-Instruct** (Yang et al., 2024). Each algorithm uses a training group of eight; the ReGRPO cohort is further split into two sub-groups of four. On GSM8K dataset (Cobbe et al., 2021), (Fig. 3) ReGRPO converges markedly faster than GRPO whether external tools (here Python is used) are enabled or not, while both reach comparable final reward. On SimuBench (Fig. 4) the advantage widens, reflecting the dataset's high interaction frequency and structured reference answers (block lists and connection details rather than single numbers).

**Two-stage curriculum.** Stage 1 trains on systems with $\leq$ 12 blocks, Stage 2 on larger models (> 12 blocks) while explicitly encouraging planning and reflection. During Stage 2, we add 200 augmentation examples that contain only a Simulink model without document; the agent must first generate a textual summary, then rebuild the model from that summary. This raises the convergence plateau (Fig. 4c). Skipping Stage 1 slows learning and lowers the final score (Fig. 4b).

**Benchmark comparison.** Table 2 shows that our full pipeline (Stage 1 + Stage 2) for SimuAgent achieves the highest overall success rate (**51.89%**), surpassing every baseline on both small and large systems. It consistently outperforms Chain-of-Thought (CoT), Retrieval-Augmented Generation (RAG) and supervised fine-tuning (SFT) across all metrics, underscoring both the value of SimuAgent and the effectiveness of the ReGRPO-based training framework.

Among pretrained baselines, GPT-4o demonstrates strong performance with its XML-based approach reaching 50.45%—narrowing the gap to just 1.44% behind SimuAgent. Notably, GPT-4o excels in modification tasks for small systems (67.29%) but still lags significantly in large system creation (8.50% vs. SimuAgent's 11.71%) and QA tasks (46.21% vs. 54.50%), highlighting the benefits of domain-specific training and tool use.

**Ablation studies.** Ablation studies reveal the critical importance of both training stages: using only Stage 1 achieves 46.55% ($-5.34$ points), while only Stage 2 reaches 44.38% ($-7.51$ points). The complementary nature of the stages is evident—Stage 1 establishes strong foundations on smaller systems, while Stage 2 enables better generalization to complex, large-scale models. With-

out any training, SimuAgent performs comparably to the base Qwen-2.5-7B model (25.29% vs. 26.57%), confirming that the performance gains stem from our training methodology rather than architectural changes.

### 4.3    GENERALIZATION AND CROSS-PLATFORM TRANSFER

We conducted experiments that demonstrate (1) the general effectiveness of ReGRPO over GRPO across diverse tasks and (2) SimuAgent's ability to transfer learned modeling principles to other platforms. Table 3 presents these results, validating both the domain-agnostic nature of our approach and the broad applicability of the reflection mechanism.

**ReGRPO's consistent advantages.**    Across all evaluated tasks, ReGRPO outperforms GRPO, confirming that our reflection mechanism extends beyond Simulink modeling. The improvements are modest but consistent in general reasoning (MMLU: +0.83%, BBH: +0.69%) and mathematics (GSM8K: +0.90%, MATH: +0.73%). The advantage becomes more pronounced in code generation tasks—on HumanEval (Chen et al., 2021), it improves by 5.9% (reaching 61.89%), and on MBPP (Austin et al., 2022) by 4.5% (reaching 79.83%)—indicating that the reflection mechanism is particularly effective in structured, feedback-rich environments similar to Simulink modeling.

**Cross-platform modeling transfer.**    The rightmost two columns of Table 3 demonstrates SimuAgent's cross-platform capabilities. Despite being trained exclusively on Simulink data, SimuAgent achieves 33.84% accuracy on Modelica (Fritzson, 2014) and 33.04% on PSCAD (Manitoba HVDC Research Centre, 2020), representing clear improvements over the untrained baseline (28.34% and 27.67%, respectively). This transfer occurs because our Python dictionary representation naturally accommodates different modeling paradigms—Simulink's signal flow, Modelica's equation-based systems, and PSCAD's electrical networks all map to similar graph structures.

With minimal domain-specific fine-tuning (607 examples for Modelica, 730 for PSCAD), performance further rises to 44.54% and 43.49% respectively. Notably, ReGRPO maintains its advantage over GRPO even in these transfer scenarios, demonstrating that the reflection mechanism generalizes across modeling platforms. Adapting to a new environment requires only implementing API interfaces, defining a component dictionary, and building a tool library—while the core ReGRPO algorithm and plan–execute framework remain unchanged. This confirms the universality of our approach, laying a solid foundation for cross-platform applications in model-driven engineering.

## 5    CONCLUSION

In this work we introduced SimuAgent, a new LLM-driven agent for automating and assisting modeling tasks in Simulink. SimuAgent adopts a compact, LLM-friendly Python-dictionary representation that cuts token usage, boosts interpretability, and enables rapid simulation. Built on a lightweight plan–execute architecture, the agent employs staged training and the ReGRPO algorithm to tackle the sparse-reward landscape typical of long-horizon decision making.

Experiments on the newly released SimuBench benchmark show that SimuAgent converges faster and achieves higher modeling accuracy than standard RL baselines. The staged curriculum and abstract–reconstruct data augmentation further improve generalization and robustness.

Because SimuAgent can be trained and deployed entirely on-premise with modest-size models, it offers a privacy-preserving, cost-effective solution for industrial workflows. Its ability to interpret, generate, and reason about complex Simulink models paves the way for the next generation of intelligent, model-driven engineering tools.

Future work will explore richer multimodal grounding (e.g., incorporating visual information from Simulink diagrams), human-in-the-loop reward modeling, and extensions to broader graph-based design environments beyond Simulink.

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

## A    APPENDIX

This appendix provides dataset details, experimental settings, ablation results, and failure analyzes that complement the main paper.

### A.1    LLM USAGE

An LLM was used solely for light copyediting (grammar and wording) on limited portions of the manuscript. It did not contribute to the study conception, methodology, experiments, analyses, or substantive writing. All content was verified by the authors, who take full responsibility for it.

### A.2    SIMUBENCH DATASET

We release *SimuBench* (available at `https://huggingface.co/datasets/SimuAgent/SimuBench`), a large-scale benchmark expressly designed to evaluate LLMs on Simulink modeling tasks. The dataset contains 5,300 tasks spanning six system-design domains: control, mechanical, electrical, fluid, thermal, and electromagnetic.

Existing official Simulink examples are insufficient for LLM training: (i) they are limited in number; (ii) many models have deeply nested subsystems that favor direct reuse over modular construction; and (iii) they rarely include analytical questions or explicit task objectives. Although a small fraction of SimuBench tasks were inspired by official examples, the vast majority were derived from open-access academic literature, standard textbooks, or carefully reconstructed models of real-world systems such as power grids and natural-gas networks.

Each task ships with a Simulink model file, its corresponding schematic, and a set of reference question–answer pairs. Tasks fall into three categories: **Model Creation**—build a model from scratch given high-level requirements; **Model Editing**—modify or extend an existing model; **Model Question-Answering**—analyze a model and answer queries about its structure, parameters, or behavior. SimuBench is fully public and intended for both training and comprehensive evaluation.

### A.3    EXPERIMENTAL SETUP AND HYPERPARAMETERS

**Hardware and Optimizer Settings**    Experiments were conducted on a single node equipped with eight H100 GPUs. Unless stated otherwise, we used a total batch size of 512 (mini-batch 256, micro-batch 64). The maximum sequence length was 8,192 tokens. The learning rate was $1 \times 10^{-6}$, and we sampled eight responses per prompt. The KL divergence coefficient was $\beta = 0.04$ and the clip ratio $\epsilon = 0.2$. Rollouts were generated with vLLM (GPU-memory utilization 0.9), using temperature 1.0 and top-p 1.0.

To standardize tool invocation and replanning, we adopt the prompting template summarized in Table 4.

**Reward Design**    For question-answering tasks, the reward combines answer correctness and output-format compliance. For model-generation tasks, additional terms evaluate (i) structural and functional similarity to a reference, (ii) completeness and executability, and (iii) successful tool usage. Rewards are min–max normalized to the range $[0, 1]$.

### A.4    HYPERPARAMETER ABLATION STUDY

We conducted an extensive ablation study to determine how each key hyper-parameter influences SimuAgent's accuracy on SimuBench. Table 5 reports the full sweep, obtained with the corrected scorer. Under the economical setting of group size $G = 8$, learning rate LR $= 10^{-6}$ and $\beta = 0.04$, the default configuration delivers the best cost–benefit balance.

- **Group size.** Shrinking the group to $G = 4$ amplifies gradient variance and lowers the overall average by –1.78%; enlarging it to $G = 16$ brings only a modest +0.25% gain.
- **Adaptive reflection.** Removing reflection drops performance by 3.05%, while forcing reflection every turn is even worse (–5.64%).

Table 4: Templates for Prompting Tool Use and Replanning. The placeholder `tool_descriptions` will be replaced with descriptions of the available tools.

---

**Prompt Template for Tool Invocation**

---

Think step-by-step inside `<think>...</think>` tags. After thinking, choose ONE of the following:
  - Call a tool inside `<tool>...</tool>` tags
  - Execute Python inside `<python>...</python>` tags
  - Provide the final answer inside `<answer>... </answer>` tags
Available tools are listed in `tool_descriptions`.
To call a tool, write a JSON object inside `<tool>` tags with:
  - "name": the tool's name
  - "args": the arguments to pass to the tool
The tool's output will appear inside `<result>` tags. Feel free to invoke tools multiple times if needed.

---

**Prompt Template for Replanning Decisions**

---

Based on the current state, decide on the most appropriate next action:
1. If the plan requires refinement, return `<tool>{"name": "plan", "args": {"plan_list": [...]}}</tool>` with the updated steps.
2. If the next step in the current plan should be executed immediately, return `<answer>Continue</answer>`.
3. If the **entire task** is complete, return `<answer>Finish</answer>`.

---

- **Two-stage curriculum.** Capping Stage 1 at $\leq 8$ blocks trims 2.31% off the mean score; skipping Stage 1 entirely costs 7.51%.
- **Reward shaping.** "Correctness-Only" lowers the average by 5.67 points, with the largest drops in creation and modification.
- **LR–KL schedule.** A higher LR of $10^{-5}$ with $\beta = 0.07$ nearly matches the default, but too small an LR ($10^{-7}$, $\beta = 0.01$) starves learning and an aggressive LR ($10^{-4}$, $\beta = 0.15$) hurts creation despite decent QA.
- **LoRA.** For LoRA, we adopt rank $r = 32$, scaling factor $\alpha = 64$, and a learning rate of $5 \times 10^{-6}$; this configuration retains $\sim$97.5% of the full fine-tune performance while remaining parameter-efficient.
- **Model scale.** Performance scales with model size: Qwen2.5-3B averages 38.69%, while Qwen2.5-14B attains 52.78% (highest overall accuracy).

These results confirm that the two-stage curriculum, ReGRPO with adaptive reflection, and structurally informed rewards dominate performance, and remain robust across model scales, optimization settings and tuning strategies.

We also compare the runtime of GRPO and ReGRPO to assess the overhead introduced by reflection; see Figure 5. ReGRPO is slower in absolute wall-clock time per rollout, but both methods scale similarly with the number of turns, and the relative overhead shrinks from about 13.5% at one turn to about 7.2% at eight turns, making the added cost increasingly negligible for longer rollouts.

A.5 FAILURE ANALYSIS ON SIMUBENCH

We conducted a comprehensive examination of 1,400 test tasks in SimuBench, systematically analyzing the 673 cases where SimuAgent failed on the first attempt. As summarized in Table 6, our analysis identifies five primary failure patterns, which collectively account for nearly all failed cases. These failure patterns exhibit strong correlation with task complexity. Large systems (>12 blocks), cross-domain editing tasks requiring adherence to conservation laws, and compound operations demanding more than five planning steps show significantly higher failure rates. These scenarios

Table 5: Ablation results for SimuAgent on SimuBench. Values are accuracy (%), higher is better. Overall best scores are in **bold**; best scores for the 7B model are underlined.

| Config. | Key Change | Create (%) | Modify (%) | QA (%) | Avg. (%) |
|---|---|---|---|---|---|
| Default | – | 30.06 | 55.11 | 64.11 | 51.89 |
| (A) $G=4$ | Smaller group size | 29.95 | 52.23 | 62.33 | 50.11 |
| (B) $G=16$ | Larger group size | 30.94 | 54.96 | 64.35 | 52.14 |
| (C) No Reflection | GRPO only | 27.37 | 54.66 | 58.02 | 48.84 |
| (D) Always Reflect | $p=1.0$ | 27.58 | 48.20 | 57.56 | 46.25 |
| (E) Fixed $p=0.5$ | Fixed reflection probability | 27.97 | 54.45 | 60.87 | 49.93 |
| (F) Stage $1 \leq 8$ | Smaller Stage 1 threshold | 28.96 | 52.30 | 61.48 | 49.58 |
| (G) No Curriculum | Skip Stage 1 | 23.96 | 46.56 | 56.74 | 44.38 |
| (H) Correctness Only | Remove format/tool-use rewards | 26.67 | 48.06 | 58.29 | 46.22 |
| (I) No Augmentation | Remove abstract-reconstruct | 29.84 | 52.19 | 63.34 | 50.43 |
| (J) LR=1e-7, $\beta$=0.01 | Lower LR & KL coefficient | 29.24 | 52.30 | 57.25 | 48.14 |
| (K) LR=1e-5, $\beta$=0.07 | Higher LR & KL coefficient | 29.60 | 55.47 | 63.86 | 51.82 |
| (L) LR=1e-4, $\beta$=0.15 | Highest LR & KL coefficient | 28.21 | 53.89 | 63.81 | 50.83 |
| (M) LoRA ($r$=32) | Parameter-efficient fine-tune | 29.74 | 54.28 | 61.66 | 50.61 |
| (N) Qwen2.5-3B | Smaller model | 20.52 | 41.00 | 49.27 | 38.69 |
| (O) Qwen2.5-14B | Larger model | **30.65** | **56.08** | **65.13** | **52.78** |

Table 6: Failure Types in SimuBench and Their Root Causes

| Failure Type | Typical Symptoms | Root Causes |
|---|---|---|
| **Topology / Connection Errors (33.9%)** | • Incorrect port naming
• Port domain mismatch
• Line direction wrong | • Limited grasp of physical domain constraints
• Python validator misses complex Simscape rules
• Complex multi-domain port conventions |
| **Block Selection / Implementation Errors (28.5%)** | • Use of similarly named incorrect blocks
• Incorrect library path hierarchy
• Missing Solver Configuration block | • Confusion from similar names across libraries
• Deep, sparsely documented library hierarchies
• Limited physical system knowledge |
| **Parameter Omission or Errors (17.7%)** | • Missing required parameter fields
• Parameter name misspellings
• Computed fields left empty | • Overlooked entries in large dictionaries
• Cross-block dependencies exceed reasoning span
• Inconsistent parameter conventions |
| **Premature Task Termination (12.5%)** | • Halting after tool errors
• Invoking `Finish` before task is done
• Misjudging completion status | • Weak self-assessment of progress
• Limited error recovery strategies
• Insufficient replanning capabilities |
| **Context Length Exceeded (7.3%)** | • Token limit reached mid-task
• Incomplete multi-stage plans
• Partial system generation | • Inadequate context window for large systems
• Verbose reasoning traces and representations
• Inefficient token allocation |

challenge the agent's capacity for port alignment, hierarchical library navigation, bulk parameter specification, and dynamic replanning.

## A.6 EXTENDED RESULTS AND CASE STUDIES

To complement the quantitative ablations, Tables 7–12 present six case studies spanning model editing and creation. The two GPT-4o failures highlight common pitfalls—library-path mismatches, invalid object references, and missing configuration—whereas SimuAgent's four cases demonstrate tool-aware block discovery, port alignment, and error recovery that yield executable models.

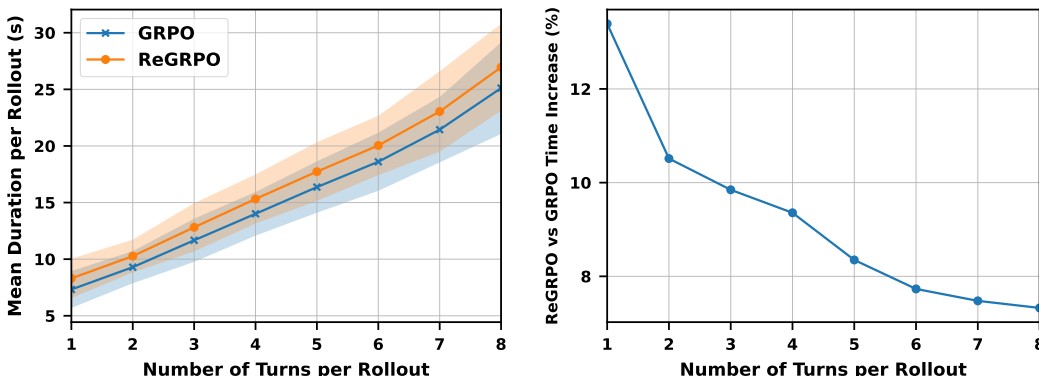

Figure 5: Runtime comparison between GRPO and ReGRPO. (a) Mean wall-clock duration per rollout (no-tool setting). ReGRPO is consistently slower than GRPO, yet both methods exhibit similar scaling with the number of turns. (b) Percentage runtime overhead of ReGRPO relative to GRPO. Although ReGRPO incurs a noticeable slowdown for short rollouts, the relative overhead steadily diminishes—from about 13.5% at one turn to about 7.2% at eight turns—indicating that its extra cost becomes less significant as rollout length (and thus typical tool-calling scenarios) increases.

Table 7: GPT-4o case study 1 (failed).

**Question**: Add a Temperature Sensor between the Fuel Cell Stack and the T_meas port of the Pump Control.

**GPT-4o**:

*Read from Image:*

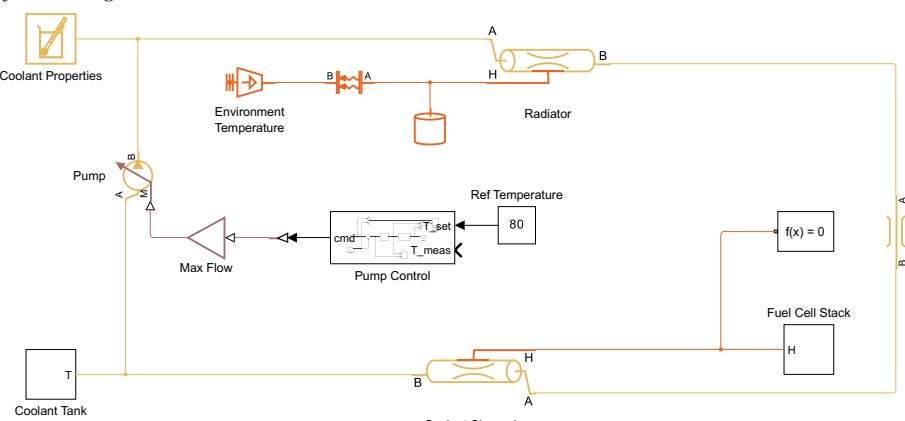

<think> Here is the MATLAB code to programmatically add a Temperature Sensor between the Fuel Cell Stack and the T_meas input of the Pump Control in the open model... </think>
<answer> model = 'CoolingSystem'; open_system(model);
sensorPos = [800 450 850 500]; converterPos = [900 450 950 500];
add_block('simscape/Foundation/Temperature Sensors/Temperature Sensor', [model '/Temperature Sensor'], 'Position', sensorPos);
add_block('simscape/Utilities/PS-Simulink Converter', [model '/PS-Simulink'], 'Position', converterPos);
add_block('simscape/Foundation/Thermal/Thermal Reference', [model '/Thermal Reference'], 'Position', [800 520 850 550]);
add_line(model, 'Fuel Cell Stack/H', 'Temperature Sensor/H', 'autorouting', 'on');
add_line(model, 'Thermal Reference/R', 'Temperature Sensor/R', 'autorouting', 'on');
add_line(model, 'Temperature Sensor/S', 'PS-Simulink/1', 'autorouting', 'on');
add_line(model, 'PS-Simulink/1', 'Pump Control/T_meas', 'autorouting', 'on'); </answer>

*Simulink Error: There is no block named 'simscape/Foundation/Temperature Sensors/Temperature Sensor'*

Table 8: GPT-4o case study 2 (failed).

**Question**: Add a Temperature Sensor between the Fuel Cell Stack and the T_meas port of the Pump Control.

**GPT-4o**:
*Read from XML:*
<?xml version="1.0" encoding="utf-8"?>
<System>
  <P Name="Location">[-1, -8, 1921, 1041]</P>
  <P Name="Open">on</P>
  <P Name="ZoomFactor">241</P>
  <P Name="ReportName">simulink-default.rpt</P>
  <P Name="SIDHighWatermark">49</P>
  <Block BlockType="Reference" Name="Convective Heat
Transfer" SID="2">
    <PortCounts lconn="1" rconn="1"/>
    <P Name="Position">[515, 226, 555, 254]</P>
    <P Name="ZOrder">230</P>
    <P Name="BlockMirror">on</P>
    <P Name="LibraryVersion">24001000.1</P>
    <P Name="SourceBlock">fl_lib/Thermal/Thermal Elements/Convective Heat
Transfer</P>
    ...... **819 lines, 20,231 tokens**
  <Line>
    <P Name="ZOrder">10</P>
    <P Name="Src">17#out:1</P>
    <P Name="Dst">28#in:1</P>
  </Line>
  <Line>
    <P Name="ZOrder">11</P>
    <P Name="Src">29#out:1</P>
    <P Name="Dst">17#in:1</P>
  </Line>
</System>

<think> To insert a temperature sensor between the Fuel Cell Stack and the T_meas input of the Pump Control block in the Simulink model named CoolingSystem, I can use the following MATLAB script: ... </think>

<answer> open_system('CoolingSystem');
sensorPath = 'CoolingSystem/Temperature Sensor'; fuelCellPath = 'CoolingSystem/Fuel Cell Stack'; pumpCtrlPath = 'CoolingSystem/Pump Control';
add_block('fl_lib/Thermal/Thermal Sensors/Temperature Sensor', ... sensorPath, ... 'Position', [790, 440, 830, 470]);
delete_line('CoolingSystem', 'Fuel Cell Stack/1', 'Pump Control/2');
add_line('CoolingSystem', 'Fuel Cell Stack/1', 'Temperature Sensor/L', 'autorouting','on');
add_line('CoolingSystem', 'Temperature Sensor/R', 'Pump Control/2', 'autorouting','on');
</answer>

*Simulink Error: Invalid Simulink object name: 'Fuel Cell Stack/1'.*

Table 9: SimuAgent case study 1 (successful).

**Question**: Add a Temperature Sensor between the Fuel Cell Stack and the T_meas port of the Pump Control.

**SimuAgent**:

*Read from Python:*

sys_dict = {"Blocks": {"Convective Heat Transfer": {"Type": "Convective Heat Transfer"}, "Coolant Channels": {"Type": "Pipe (TL)"}, "Coolant Properties": {"Type": "Thermal Liquid Settings (TL)"}, "Coolant Tank": {"Type": "Coolant Tank"}, "Environment Temperature": {"Type": "Temperature Source"}... }, "Connections": [{"Src": "Convective Heat Transfer/LConn1", "Dst": "Radiator/LConn2"}, {"Src": "Thermal Mass/LConn1", "Dst": "Radiator/LConn2"}, ...]}

<think> The user wants to add a temperature sensor. I need to identify the specific block for a temperature sensor. The sensor should be placed between "Fuel Cell Stack" and the "T_meas" port of "Pump Control". I should first search for the "Temperature Sensor" block to get its details ... </think>

<tool> {"name": "search_blocks", "args": {"query_list": ["Temperature Sensor"]}} </tool>

<result> "Temperature Sensor"
- Block: Temperature Sensor
- Library: Simscape / Foundation Library / Thermal / Thermal Sensors
- Description: The Temperature Sensor block represents an ideal temperature sensor, that is, a device that measures either the absolute temperature at a node, or the temperature difference between two nodes, without drawing any heat.
Connections A and B are thermal conserving ports that connect to the two points where temperature is being monitored. Port T is a physical signal port that outputs the temperature difference value. ... </result>

<think> Okay, I have the "Temperature Sensor" block details. It has thermal ports (A, B) and a physical signal output (T). To measure temperature from the "Fuel Cell Stack", I'll connect one thermal port of the sensor (let's assume port A, which will be LConn1 in SimuAgent's abstraction) to an output of the "Fuel Cell Stack". ... </think>

<python> sys_dict["Blocks"]["Temperature Sensor"] = "Type": "Temperature Sensor", "Temperature_measure": "Absolute"
sys_dict["Blocks"]["PS-Simulink Converter"] = "Type": "PS-Simulink Converter"
sys_dict["Connections"].append({"Src": "Fuel Cell Stack/LConn1", "Dst": "Temperature Sensor/LConn1"})
sys_dict["Connections"].append({"Src": "Temperature Sensor/RConn2", "Dst": "PS-Simulink Converter/LConn1"})
sys_dict["Connections"].append({"Src": "PS-Simulink Converter/1", "Dst": "Pump Control/2"})
</python>

<result> Python execution completed. No errors, and no result returned. </result>

*Based on the current state, decide on the most appropriate next action: ...*

<answer> Finish </answer>

*Convert sys_dict_detailed into MATLAB Commands*:

add_block('fl_lib/Thermal/Thermal Sensors/Temperature Sensor', 'CoolingSystem/Temperature Sensor', 'temperature_measure', "false");
add_block('nesl_utility/PS-Simulink Converter', 'CoolingSystem/PS-Simulink Converter');
add_line(model, 'Fuel Cell Stack/LConn1', 'Temperature Sensor/LConn1', 'autorouting', 'on');
add_line(model, 'Temperature Sensor/RConn2', 'PS-Simulink Converter/LConn1', 'autorouting', 'on');
add_line(model, 'PS-Simulink Converter/1', 'Pump Control/2', 'autorouting', 'on');
% Simulink.BlockDiagram.arrangeSystem(gcs, FullLayout='true')

**Simulink**:

Table 10: SimuAgent case study 2 (successful).

**Abstract Question**: Generate a technical abstract (under 200 words) for the system provided. Detail its essential blocks, their interconnections, and any critical parameter settings.

**SimuAgent**:

*Read from Python dictionary*:

sys_dict = {"Blocks": {"Solver Configuration": "Type": "Solver Configuration", "AC Voltage Source": {"Type": "AC Voltage Source"}, "Eref": "Type": "Electrical Reference", ...}, "Connections": [{"Src": "AC Voltage Source/-", "Dst": "Electrical Reference/V"}, ...]}

<think> The sys_dict contains the blocks and their connections. I need to identify the main functional components (like sources, passive elements, active elements like op-amps, sensors, and scopes) and describe how they are connected. ... </think>

<answer> The circuit features an operational amplifier (op-amp) configured as a differentiator or low-pass filter. An AC voltage source is connected between ground and a node. This node is also connected to one side of a capacitor (C) and to the positive terminal of a voltage sensor, whose output is labeled "Vin". The other side of capacitor C connects to the inverting input (-) of the op-amp.

The non-inverting input (+) of the op-amp is grounded. The op-amp's output is connected to one end of a resistor (R) and to the positive terminal of a second voltage sensor, whose output is labeled "Vout". The other end of resistor R provides feedback to the op-amp's inverting input. Both voltage sensors have their negative terminals grounded.

A Solver Configuration block is connected to the node where the "Vin" signal is measured (between the AC source and capacitor C), and its other connection is to ground. The signals "Vin" and "Vout" are then fed into a Mux, which outputs to a Scope block labeled "Voltages" for waveform display. </answer>

**Reconstruct Question**: Create the system as described in the provided abstract: The circuit features an operational amplifier (op-amp) configured as a differentiator or low-pass filter. ...

**SimuAgent**:

<think> The user wants to construct a system based on a textual description. I need to parse this description to identify all the necessary circuit components (op-amp, AC voltage source, ground, capacitor, resistor, voltage sensors, Solver Configuration). I will then search for these blocks to get their library paths and confirm their existence. ... </think>

<tool> {"name": "search_blocks", "args": {"query_list": ["op-amp", "AC voltage source", "ground", "capacitor", "resistor", "voltage sensors"]}} </tool>

<result> "op-amp"
- Block: Op-Amp
- Library: Simscape / Foundation Library / Electrical / Electrical Elements
- Description: The Op-Amp block models an ideal operational amplifier (op-amp). The figure shows the implementation schematic. ... "AC voltage source" ... "capacitor" ... </result>

<think> I have the search results for the primary components. Now, I need to process the full textual description again, this time to define each block instance (giving them names if not specified or using defaults) and then to establish all the connections as described. I'll need to pay attention to specific terminals (e.g., op-amp inverting/non-inverting inputs, sensor positive/negative terminals). ... </think>

<python> sys_dict = {"Blocks": {"AC Voltage Source": {"Type": "AC Voltage Source"}, "Capacitor": "Type": "Capacitor", "Eref": "Type": "Electrical Reference", ...}, "Connections": [{"Src": "AC Voltage Source/-", "Dst": "Electrical Reference/V"}, ...]} </python>

<result> Python execution completed. No errors, and no result returned. </result>

- - - - - - - - - - - - - - - - - - - - - - - - - - - - - - - -
*Based on the current state, decide on the most appropriate next action: ...*

<answer> Finish </answer>

*Convert sys_dict_detailed into MATLAB Commands*:

add_block('fl_lib/Electrical/Electrical Sources/AC Voltage Source','untitled/AC Voltage Source');
add_block('fl_lib/Electrical/Electrical Elements/Capacitor','untitled/Capacitor'); ...

**Simulink**:

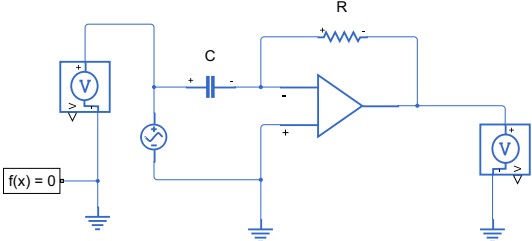

Table 11: SimuAgent case study 3 (failed).

**Question**: Create an ideal AC transformer plus full-wave bridge rectifier.

**SimuAgent**:

<think> The user wants a system with an "ideal AC transformer" and a "full-wave bridge rectifier." This is a common power electronics circuit. I need to break this down into logical construction steps. First, the AC input and transformer stage. </think>

<tool> "name": "plan", "args": {"plan_list": ["Establish the AC voltage source that will serve as the input. Following this, model the ideal transformer, connecting its primary winding to the AC source.", "Building the full-wave bridge rectifier itself. This requires correctly placing and orienting the four diodes (Diode 1, Diode 2, Diode 3, and Diode 4) to form the bridge configuration.", "Model the output section of the circuit." </tool>

<result> subsys_dict_1 = {"Blocks": { "AC Voltage Source": { "Type": "AC Voltage Source" }, "Ideal_Transformer": { "Type": "Ideal Transformer" }, "ERef_T1": { "Type": "Electrical Reference" }, "ERef_T2": { "Type": "Electrical Reference" },...},... } </result>

*Based on the current state, decide on the most appropriate next action: 1. If the plan requires refinement, return <tool>{"name": "plan", "args": {"plan_list": [...]}}</tool> with the updated steps. 2. If the next step in the current plan should be executed immediately, return <answer>Continue</answer>. 3. If the **entire task** is complete, return <answer>Finish</answer>.*

<think> The plan tool has successfully generated subsys_dict_1, which covers the AC source and the ideal transformer, as per the first part of my initial plan. The overall task also requires the full-wave bridge rectifier and the output section. ... </think>

<answer> continue </answer>

<result> subsys_dict_2 = {"Blocks": { "AC Voltage Source": { "Type": "AC Voltage Source" }, "C": { "Type": "Capacitor" }, "Diode 1": { "Type": "Diode" }, "Diode 2": { "Type": "Diode" },...}, "Connections": [ {"Src": "Diode 3/RConn1", "Dst": "Diode 1/LConn1" }, {"Src": "Diode 3/RConn1", "Dst": "Ideal Transformer/RConn1" }, {"Src": "AC Voltage Source/LConn1", "Dst": "Ideal Transformer/LConn1" }, {"Src": "PS-Simulink Converter/LConn1", "Dst": "Solver Configuration/RConn1" },... ] } </result>

<think> I now have subsys_dict_1 (AC source and transformer) and subsys_dict_2 (rectifier, load, other components). To get the complete system model as requested, I need to merge these two partial system dictionaries. This involves combining the "Blocks" dictionaries and the "Connections" lists from both. ... </think>

<python> sys_dict = {"Blocks": { "AC Voltage Source": { "Type": "AC Voltage Source" }, "C": { "Type": "Capacitor" }, "Diode 1": { "Type": "Diode" }, "Diode 2": { "Type": "Diode" },...}, "Connections": [ {"Src": "Diode 3/RConn1", "Dst": "Diode 1/LConn1" }, {"Src": "Diode 3/RConn1", "Dst": "Ideal Transformer/RConn1" }, {"Src": "AC Voltage Source/LConn1", "Dst": "Ideal Transformer/LConn1" }, {"Src": "PS-Simulink Converter/LConn1", "Dst": "Solver Configuration/RConn1" },... ] } </python>

<result> Python execution completed. No errors, and no result returned. </result>

*Based on the current state, decide on the most appropriate next action: ...*

<answer> Finish </answer>

*Convert sys_dict into MATLAB Commands*:

... add_line('untitled','Diode_1/1','C/1'); ...

*Simulink Error: Invalid Simulink object name: 'Diode_1/1'.*

**Simulink**:

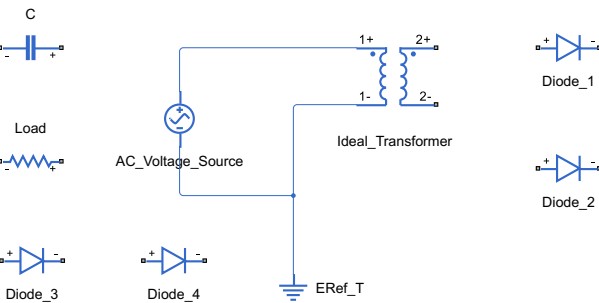

Table 12: SimuAgent case study 4 (successful).

**Question**: Design a system where a DC motor is mechanically connected to a Permanent Magnet Synchronous Generator (PMSG), which converts mechanical energy into electrical energy. The PMSG is connected to a converter, which then supplies a DC load via a DCDC.

**SimuAgent**:

`<think>` I need to search for the library blocks corresponding to the components mentioned in the user query: "DC motor", "Permanent Magnet Synchronous Generator", "ACDC converter", and "DCDC". ... `</think>`

`<tool>` {"name": "search_blocks", "args": {"query_list": ["DC motor", "Permanent Magnet Synchronous Generator", "ACDC converter", "DCDC"], "lib": "sps_lib"}} `</tool>`

`<result>` "DC Motor"
- Block: DC Machine
- Library: Simscape / Electrical / Specialized Power Systems / Electrical Machines
- Description: Implements a wound-field or permanent magnet DC machine ...

"PMSG"
- Block: Permanent Magnet Synchronous Machine
- Library: Simscape / Electrical / Specialized Power Systems / Electrical Machines
- Description: Models a three-phase or five-phase permanent magnet synchronous machine ...

- Block: Simplified Synchronous Machine
- Library: Simscape / Electrical / Electromechanical / Synchronous
- Description: Models the electrical and mechanical characteristics of a simplified synchronous machine ...

"ACDC converter" ... `</result>`

`<think>` I need to list all the necessary blocks and assign them a "Type". For "DC motor", the search returned "DC Machine", so I'll use {"Type": "DC Machine"}. For "PMSG", the search returned "Permanent Magnet Synchronous Machine", so I'll use {"Type": "Permanent Magnet Synchronous Generator"}. For "AC-DC Converter", ... `</think>`

`<python>` sys_dict = {"Blocks": {"DC Machine": {"Type": "DC Machine"}, "Permanent Magnet Synchronous Generator": {"Type": "Permanent Magnet Synchronous Generator"}, "AC-DC Converter": {"Type": "Three-Level NPC Converter"}, "DC-DC Converter": {"Type": "DC Machine"}, "DC Load": {"Type": "DC Machine"}}, "Connections": [{"Src": "DC Machine/1", "Dst": "Permanent Magnet Synchronous Generator/1"}, {"Src": "Permanent Magnet Synchronous Generator/(LConn1,LConn2,LConn3)", "Dst": "AC-DC Converter/(LConn1,LConn2,LConn3)"}, ...]} `</python>`

`<result>` Python execution completed. No errors, and no result returned. `</result>`

*Based on the current state, decide on the most appropriate next action: ...*

`<answer>` Finish `</answer>`

*Convert sys_dict into MATLAB Commands*:

add_block('spsDCMachineLib/DC Machine','untitled/DC Machine');

add_block('spsPermanentMagnetSynchronousMachineLib/Permanent Magnet Synchronous Machine', 'untitled/Permanent Magnet Synchronous Machine');

...

add_line('untitled','DC-DC Converter/RConn1','DC Load/LConn1');

add_line('untitled','DC Load/RConn1','DC-DC Converter/LConn2');

**Simulink**:

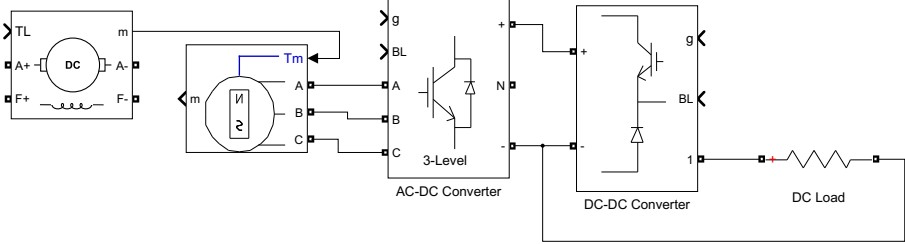