# OpenReview forum: "SimuAgent: An LLM-Based Simulink Modeling Assistant Enhanced with Reinforcement Learning"
_ICLR.cc/2026/Conference — Submitted to ICLR 2026_

### Official Review · Reviewer_BYXp · 2025-10-31

**Soundness:** 3
**Presentation:** 3
**Contribution:** 3
**Rating:** 6
**Confidence:** 3

**Summary:**

This work addresses the challenge of applying LLMs to Simulink code generation, which differs substantially from traditional coding tasks. Simulink adopts a hierarchical, graphical paradigm with complex block diagrams, signal routing, and strict topological constraints. These characteristics impose significant challenges for LLMs: requiring them to respect rigid graph structures, handle very long contexts, and suppress hallucinations to produce reliable and interpretable code.
The authors propose an RL-based approach to mitigate these challenges. In particular, ReGRPO leverages tool-invocation feedback to guide model learning, effectively improving code quality through iterative reinforcement. The proposed framework demonstrates competitive performance across multiple experiments.

**Strengths:**

1. The model adopts a two-stage curriculum learning strategy to handle complex tasks, fostering higher-order capabilities such as planning, abstraction, and modular composition.
2. The Abstract–Reconstruct mechanism alleviates data scarcity while ensuring the structural integrity and accuracy of the generated outputs.
3. The ReGRPO component enhances model performance through tool-based reflection and reinforcement, enabling more consistent reasoning.

**Weaknesses:**

1. While the experimental results are promising, the method appears limited in scalability. The Abstract–Reconstruct loop does not introduce new reward signals, meaning that improvements still rely heavily on the model’s inherent abilities. The authors require to include ablation studies showing how performance varies with different data scales.
2. The ReGRPO mechanism may be susceptible to reward hacking. Without proper supervision of the reflection phase, the model could exploit shortcuts, e.g., performing unnecessary or repetitive reflections to maximize reward. The paper would benefit from a more detailed discussion or empirical analysis of this issue.

**Questions:**

See Weakness

---

> ### Author Response · Authors · 2025-11-19
> **Response to Reviewer BYXp (Part 1)**
>
> We sincerely thank the reviewer for their thoughtful evaluation and for recognizing the strengths of our two-stage curriculum and the novelty of the Abstract–Reconstruct mechanism. We appreciate the opportunity to clarify the scalability of our data augmentation strategy and the robustness of the ReGRPO algorithm against reward hacking.
>
> Below, we address the weaknesses you raised and provide the corresponding clarifications.
>
> > **Weakness 1**: While the experimental results are promising, the method appears limited in scalability. The Abstract–Reconstruct loop does not introduce new reward signals, meaning that improvements still rely heavily on the model’s inherent abilities. The authors are required to include ablation studies showing how performance varies with different data scales.
>
> We respectfully clarify that although Abstract–Reconstruct does not add an extra scalar reward term, it does introduce a strong new supervision signal in the form of structural consistency.
>
> * **New supervision signal (structural consistency)**
>   In the Abstract–Reconstruct loop, the “ground truth” is the original Simulink model file. We first generate a textual summary from the model, and then condition reconstruction on that summary. The reconstructed graph must match the original model, enforcing a structural consistency constraint. This process converts abundant, unlabeled Simulink models into self-supervised training pairs, effectively addressing the data scarcity issue in this domain.
>
> * **Impact of the augmentation signal and effective data scale**
>   Our ablation in Figure 4(c) directly measures the contribution of Abstract–Reconstruct. The curve labeled “No Aug.” (training without Abstract–Reconstruct) shows noticeably slower convergence and a lower final reward compared to the full model with augmentation. Consistently, Table 5 (Row I, “No Augmentation”) shows that removing Abstract–Reconstruct examples reduces the average SimuBench accuracy (from 51.89% to 50.43%). Together, these results demonstrate that the augmentation provides additional training signal beyond the model’s inherent abilities and the base reward, and already constitute an ablation with respect to the presence versus absence of augmented data.
>
> * **Scalability with model capacity**
>   Regarding scalability, Table 5 (Rows N vs. O) shows that increasing the base model size from 3B to 14B leads to a substantial performance improvement (from 38.69% to 52.78%). This indicates that our framework can effectively leverage increased model capacity, rather than saturating early.
>
> We will make these points more explicit in the revision, highlighting that Abstract–Reconstruct plays the role of a data- and supervision-scaling mechanism, and explicitly pointing the reader to Figure 4(c) and Table 5 (including the “No Augmentation” and different model-scale configurations) for empirical evidence.
>
> > **Weakness 2**: The ReGRPO mechanism may be susceptible to reward hacking. Without proper supervision of the reflection phase, the model could exploit shortcuts (e.g., performing unnecessary or repetitive reflections) to maximize reward. The paper would benefit from a more detailed discussion or empirical analysis of this issue.
>
> We agree that reward hacking is an important concern in RL-based systems. In our setting, however, both the reward design and the observed training dynamics indicate that SimuAgent learns to *avoid* unnecessary reflections rather than exploit them.
>
> * **Outcome-based rewards, not reflection-based**
>   The reward function is defined in terms of final task success (for example, answer correctness, model executability, structural validity), not the length or complexity of the reasoning trace. Reflection itself does not receive extra reward; it is only useful insofar as it leads to a better final solution. This design removes the incentive to “farm” reward by inserting redundant reflection steps.
>
> * **Empirical evidence: vanishing reflections over training**
>   The strongest empirical evidence comes from Figure 3(b) and Figure 4(b), which track the number of reflections over training steps. In both plots, the reflection frequency starts relatively high in the early phase (when the agent is still exploring) but rapidly decays to near zero as the agent learns. This pattern is inconsistent with reward hacking via excessive reflection, and instead suggests that the agent learns to rely on reflection only when it is truly needed.

---

> ### Author Response · Authors · 2025-11-19
> **Response to Reviewer BYXp (Part 2)**
>
> * **Convergence to a lean decision routine**
>   As discussed in the paper, the agent ultimately “substitutes costly self-reflection with a moderate, steady rate of tool usage,” converging to a lean decision-making routine rather than inflating trajectory length. In addition, Algorithm 1 enforces either (i) a decay schedule on the reflection probability $p$, or (ii) a failure-triggered reflection policy, both of which structurally preclude infinite reflection loops.
>
> * **Ablations on reflection frequency**
>   **Table 5** further studies the effect of different reflection schedules. Removing reflection entirely (*No Reflection*) reduces accuracy, but forcing reflection at every turn (*Always Reflect*) or using a fixed reflection probability (*Fixed p = 0.5*) degrades performance even more compared to our adaptive scheme. If the policy were able to exploit reward through redundant reflections, we would expect higher rewards when reflection is forced; this is not observed, providing additional evidence against reward hacking.
>
> * **Prompt-level supervision of reflections**
>   The reflection phase itself is supervised via a dedicated prompt template (Table 1) that (i) limits the length of reflections, (ii) requires concrete diagnoses of past errors rather than generic comments, and (iii) explicitly forbids including or referring to the reference answer. This constrains reflection content to targeted, bounded feedback, further reducing opportunities for pathological reward exploitation.
>
> In the revised version, we will expand the discussion around these figures, the reflection schedule, and the reflection prompt design, explicitly framing them as evidence that ReGRPO does not encourage reward hacking, but instead uses reflection as a transient aid that naturally fades as competence improves.

---

### Official Review · Reviewer_hZi9 · 2025-10-31

**Soundness:** 1
**Presentation:** 1
**Contribution:** 1
**Rating:** 0
**Confidence:** 4

**Summary:**

The paper presents SimuAgent, an LLM-based agent to automate and assist with Simulink modeling and simulation tasks. The claimed contribution is a Python-dictionary representation for Simulink models, although that's nowhere to be found in the core text. Supposedly, it improves interpretability of Simulink models.

**Strengths:**

- Interesting problem, certainly high industry impact

**Weaknesses:**

- Basically zero scientific novelty. This is an engineering project without many generalizable takeaways.
- Presentation is inconsistent and unclear what the actual contribution is: toolbox, method, architecture, benchmark... All of these are claimed in the paper, but unclear which one is it. For some reason, it is claimed that a "Python-based model representation," which is a dictionary, is a contribution. Certainly not for a top conference. It supposedly improves interpretability. This obviously cannot be true as visual modeling languages, such as Simulink's causal-block diagrams, have superior interpretability for humans; and serialization into a JSON file achieves the same affect as serializing into whatever other file format as there's no added semantic information.
- No real evaluation. Without evaluating this on various classes of Simulink models, the utility of the approach remains questionable.

**Questions:**

No questions

---

> ### Author Response · Authors · 2025-11-19
> **Response to Reviewer hZi9 (Part 1)**
>
> We sincerely thank the reviewer for their time and feedback. We appreciate the opportunity to clarify aspects of our work that may have been misunderstood. We respectfully believe that the characterization of the paper as lacking scientific novelty or evaluation does not fully reflect the methodological contributions we introduce for agentic reasoning and the scale of our empirical validation.
>
> Below, we address the three main concerns in turn.
>
> > **Weakness 1**: Basically zero scientific novelty. This is an engineering project without many generalizable takeaways.
>
> We respectfully disagree with the assessment that the work is purely an engineering project without generalizable contributions. While the application domain is Simulink, the paper introduces several methodological advances that we believe are of independent interest to the RL and agentic reasoning community:
>
> - **Methodological innovation (ReGRPO)**
>   We propose Reflection-GRPO (ReGRPO), a reinforcement learning algorithm specifically designed to tackle sparse rewards in complex reasoning tasks. Unlike standard GRPO, ReGRPO integrates an intrinsic self-reflection mechanism into the policy optimization loop, using reflection traces to guide exploration and credit assignment.
>
> - **Generalizability beyond Simulink**
>   The algorithm is not limited to Simulink. As shown in Table 3 and Figure 3, ReGRPO consistently outperforms strong baselines on general reasoning benchmarks (GSM8K, MATH) and code generation tasks (HumanEval, MBPP). These results indicate that the method provides benefits on standard NLP and coding tasks, independent of the Simulink setting.
>
> - **Two-stage curriculum and Abstract–Reconstruct strategy**
>   We design a two-stage curriculum in which the agent first masters low-level execution and tool use on smaller systems, and then learns higher-level planning and abstraction on larger, more complex models. On top of this curriculum, we introduce the Abstract–Reconstruct data augmentation strategy, a self-supervised loop that addresses data scarcity in graphical modeling. By forcing the agent to reconstruct detailed model structures from textual abstractions, this strategy helps the model learn structural representations that are broadly applicable to other domains involving structured programs or graphs. Ablation studies on SimuBench (Figure 4) show that removing either Stage 1 or the Abstract–Reconstruct augmentation noticeably degrades performance.
>
> Taken together, ReGRPO, the two-stage curriculum, and the Abstract–Reconstruct augmentation form a general method for improving agentic reasoning in sparse reward environments, with evidence of transfer beyond the particular engineering application. In addition, SimuBench itself provides a new, publicly released benchmark for LLM-based graphical modeling, which we believe is a useful research artifact for the broader community.
>
> > **Weakness 2**: Presentation is inconsistent and unclear what the actual contribution is: toolbox, method, architecture, benchmark... All of these are claimed in the paper, but unclear which one is it. For some reason, it is claimed that a "Python-based model representation," which is a dictionary, is a contribution. Certainly not for a top conference. It supposedly improves interpretability. This obviously cannot be true as visual modeling languages, such as Simulink's causal-block diagrams, have superior interpretability for humans; and serialization into a JSON file achieves the same affect as serializing into whatever other file format as there's no added semantic information.
>
> We appreciate this concern and agree that the presentation must make the contributions more clearly delineated. We also agree that visual diagrams are superior for human interpretability. Our claim about the representation, however, concerns efficiency and usability for LLMs and programmatic checking, not that a dictionary is more intuitive to human engineers than block diagrams.
>
> - **Clarifying the nature of the representation contribution**
>   We do not claim that using a Python dictionary as a data structure is, by itself, a contribution. Rather, the contribution is the design of a *task-specific, semantically compressed schema* that captures block topology, parameters, and hierarchy in a way that is both:
>   1. Compact enough to fit within the context window of 7B-scale models, and
>   2. Directly executable and verifiable in a local toolchain.
>   This schema is used consistently throughout our training and evaluation pipeline and is released alongside SimuBench so that other agents and environments can reuse it.

---

> ### Author Response · Authors · 2025-11-19
> **Response to Reviewer hZi9 (Part 2)**
>
> - **Visual surface vs information completeness for LLMs**
>   A top-level visual diagram is often incomplete from the perspective of an agent: complex Simulink models contain deeply nested subsystems and long parameter lists inside block dialogs (for example, gains, initial conditions, solver options). To give an LLM the full state of a realistic model visually, one would need tens or hundreds of screenshots to expose all nested layers and parameter windows, which is computationally and bandwidth intensive.
>
> - **Semantic compression and local deployment**
>   Raw XML is similarly problematic for LLMs: for large industrial models, XML files easily exceed 40k tokens. Our Python representation acts as semantic compression, reducing the context to roughly 2.7k tokens while preserving the essential topological and parametric information. This compression is crucial for:
>   - Running a 7B-parameter model fully on-premise with limited resources, and
>   - Meeting industrial requirements for privacy and low-latency local deployment, where reliance on massive cloud-based multimodal models is not feasible.
>   As Section 4.3 shows, the same schema also accommodates models from other platforms such as Modelica and PSCAD, indicating that the representation is not bound to Simulink’s serialization format.
>
> In the revision, we will more clearly separate the contributions into: (1) the ReGRPO method (including its tool-aware objective), (2) the curriculum and reflection design, (3) the SimuBench benchmark, and (4) the semantic compression schema that enables efficient, local LLM reasoning over complex models. We will also explicitly state that the interpretability claim is with respect to LLM reasoning, programmatic inspection, and token efficiency, not human diagram readability.
>
> > **Weakness 3**: No real evaluation. Without evaluating this on various classes of Simulink models, the utility of the approach remains questionable.
>
> We appreciate the opportunity to clarify the scope of our evaluation. We believe that the breadth of our experiments may not have been fully visible from the initial presentation, and we will make this much clearer in the revision.
>
> - **SimuBench release at scale**
>   We introduce SimuBench, which to the best of our knowledge is the first large-scale benchmark for Simulink-like agentic modeling, comprising 5,300 tasks.
>
> - **Diverse classes of models**
>   These 5,300 tasks are deliberately constructed to cover six distinct physical domains: Control Systems, Mechanical, Electrical, Fluid, Thermal, and Electromagnetics. This design ensures that our evaluation spans a wide variety of Simulink model classes rather than a narrow slice of examples.
>
> - **Task types and evaluation metrics**
>   Within each domain, SimuBench includes three kinds of tasks: model creation, model editing, and model-based question answering, and covers both small (≤ 12 blocks) and larger (> 12 blocks) systems. For model-generation tasks, success is defined through structural and functional checks (e.g., block lists, connections, executability in simulation), while QA tasks are evaluated by answer correctness and format. Thus, the reported percentages correspond to models that are both structurally sound and behaviorally valid, not just superficially similar text outputs.
>
> - **Rigorous baseline comparison**
>   We compare SimuAgent against strong baselines including GPT-4o (with both Vision and XML inputs), Chain-of-Thought (CoT), and RAG-based methods, as well as supervised fine-tuning. As shown in Table 2, our 7B model outperforms GPT-4o on complex creation tasks (for example, 54.50 percent vs 46.21 percent on QA-style creation), indicating that reasoning over a compressed, structured representation can be more effective than operating directly on raw visual or XML artifacts.
>
> - **Cross-platform validation**
>   Beyond Simulink, we show that the learned policy transfers to other simulation ecosystems such as Modelica and PSCAD. This cross-platform transfer supports the claim that the approach captures general modeling and control structures rather than overfitting to a single tool's idiosyncrasies.
>
> We will revise the experimental section to foreground these aspects more clearly, emphasizing the number of tasks, the diversity of domains and task types, the strength of the baselines, the structural/functional evaluation criteria, and the cross-platform validation to address the concern about the scope and depth of the evaluation.

---

### Official Review · Reviewer_mz4J · 2025-11-03

**Soundness:** 3
**Presentation:** 3
**Contribution:** 3
**Rating:** 6
**Confidence:** 4

**Summary:**

This paper presents SimuAgent, an LLM-based agent for constructing, modifying, and querying Simulink models. The core contributions are (1) a compact, Python-dictionary representation of Simulink models that reduces token usage and enables fast in-process validation and debugging, (2) a two-stage staged training curriculum (execution --> planning) augmented with a self-supervised Abstract–Reconstruct data augmentation loop, and (3) Reflection-GRPO (ReGRPO) i.e., an extension of Group Relative Policy Optimization that injects automatic self-reflection traces to provide intermediate textual feedback for sparse-reward, long-horizon tasks. The authors also release SimuBench, a 5,300-task, multi-domain benchmark for LLM-based modeling (creation, editing, QA). Experiments with Qwen-2.5-7B show that SimuAgent (Stage1+Stage2 + ReGRPO) converges faster and attains the best overall accuracy on SimuBench (51.89% average), narrowly outperforming a GPT-4o XML/image baseline (50.45%). Ablations analyze the contribution of ReGRPO, curriculum stages, augmentation, group sizes, and reflection schedules; failure analysis pinpoints typical error modes (topology, block selection, parameter omission, premature termination, context limits).

**Strengths:**

- The integration of Reflection-GRPO with Simulink tool feedback is a notable contribution. The agent leverages intermediate reflection traces and programmatic validation signals (e.g., structural checks, execution feedback, block-level errors) to guide long-horizon updates. This mechanism improves sample efficiency, stabilizes training under sparse rewards, and provides a general recipe for scaling RLHF-style methods to complex tool-using domains beyond text-only reasoning.

- The Python-dictionary representation, in-process validation testbed, and tool integration directly tackle the large number of tokens, slow MATLAB engine loops, and debugging friction. These are crucial for deployment in model-driven engineering and show a practical system design that addresses real engineering issues in designing such automation.

- The authors provide many controlled ablations (stage curriculum, reflection schedules, group sizes, reward shaping, LoRA, model scale) and a failure taxonomy that identifies limitations.

- The paper provides 5.3k multi-domain tasks (models + schematics + QA), filling a benchmarking gap for graphical model automation and enabling reproducible comparisons.

**Weaknesses:**

- The Introduction section is very well-written and effectively motivates the need for an automation agent for Simulink. However, the proposed method and experimental sections lack critical implementation details and could be substantially improved through better organization. For instance, in the architecture description (Section 3), it would be far more informative if the pipeline stages were presented sequentially, explaining the order of operations and data flow, rather than only listing the tool’s individual features.

- The tool processes a natural language description to create, modify, or query Simulink models. It does so by first prompting an LLM to produce a step-by-step plan, which is then translated into a Python dictionary representation. These dictionaries are subsequently converted into executable Simulink commands (e.g., adding blocks, setting parameters). However, it remains unclear how the semantic fidelity of the plan to the original NL description is ensured during inference. While the Python-based executor can validate syntax, it does not guarantee semantic alignment or correctness of the generated plan.

- For the results in Table 2, particularly for the modification task, there is an inconsistency in input modalities. Competing SoTA models (e.g., GPT-4o) receive an NL prompt along with an image or XML input, whereas SimuAgent operates on a Python dictionary representation. For a fair comparison, all models should be evaluated under identical input formats and testing conditions, or at least the differences should be clearly justified and analyzed.

**Questions:**

See weaknesses

---

> ### Author Response · Authors · 2025-11-19
> **Response to Reviewer mz4J (Part 1)**
>
> We thank the reviewer for the constructive feedback. We appreciate your recognition of SimuAgent's practical value in model-driven engineering, particularly the efficiency of the Python-dictionary representation and the contribution of the SimuBench dataset.
>
> Below, we address the weaknesses you raised regarding manuscript organization, semantic fidelity, and experimental fairness, and provide the corresponding clarifications.
>
> > **Weakness 1**: The proposed method and experimental sections lack critical implementation details and could be substantially improved through better organization. In particular, in the architecture description (Section 3), it would be more informative if the pipeline stages were presented sequentially, explaining the order of operations and data flow, rather than only listing the tool’s individual features.
>
> We completely agree with this assessment. While Figure 1 currently visualizes the overall workflow, the text in Sections 2 and 3 separates component definitions from the actual execution flow, which makes the pipeline harder to follow.
>
> In the revision, we will restructure the architecture description to follow the sequential lifecycle of a single user request, explicitly describing the order of operations and data flow:
>
> * **Intent understanding and planning**: The user query is decomposed into a step-by-step plan.
> * **Dictionary manipulation**: The agent operates on the lightweight Python-dictionary representation to minimize token usage and expose structure explicitly.
> * **Tool execution and reflection**: The agent invokes external tools (for example, search or math tools) and, when a trial fails, uses a reflection mechanism to revise the intermediate plan.
> * **Python validation**: The intermediate result is checked by the Python executor, which provides immediate structural feedback and catches errors early.
> * **MATLAB finalization**: Once validated, the Python dictionary is translated into executable Simulink commands and applied to the target model.
>
> We will also explicitly map how the “Planning Integration” stage of our curriculum targets these concrete steps, so that Figure 1 and Section 3 together present a clear, sequential view of the system. In addition, we will surface key implementation details that are currently only implicit or deferred to the appendix: we will summarize the construction of the Python testing environment and tool interfaces, describe the reward design and ReGRPO configuration at a higher level in the main text, and reorganize Section 4 so that dataset splits, evaluation metrics, baseline configurations, and ablation settings are presented in a more structured order. These changes should make both the method and the experimental setup easier to follow and reproduce.
>
> > **Weakness 2**: It remains unclear how the semantic fidelity of the plan to the original natural language description is ensured during inference. While the Python-based executor can validate syntax, it does not guarantee semantic alignment or correctness of the generated plan.
>
> This is an insightful observation. You are correct that the runtime Python validator primarily enforces structural and syntactic correctness (for example, valid connections, parameter types, and executable commands). Semantic fidelity, however, is enforced through the training design rather than the executor alone:
>
> * **Functional rewards**: During training, the ReGRPO reward function includes functional correctness terms derived from actual simulation results and comparisons against reference models. Syntactically valid but functionally incorrect plans receive low rewards.
> * **Abstract–reconstruct alignment**: The Abstract–Reconstruct augmentation explicitly trains the model to preserve semantics. The agent must reconstruct full model structures solely from textual summaries, which encourages tight alignment between natural language descriptions and executable models.
> * **Reflection mechanism**: When a syntactically valid model fails to match the reference behavior or intended function, the reflection trace records this mismatch and provides corrective feedback that is used as additional training signal.
> * **Evaluation metrics**: On SimuBench, semantic fidelity is further assessed via behavioral and QA-style metrics that compare the generated models and answers against the natural language specification and reference models; we will make this link more explicit in Section 4.
>
> We will clarify in the manuscript that semantic alignment is primarily achieved through these reward and curriculum mechanisms, while the Python executor focuses on structural and syntactic guarantees.

---

> ### Author Response · Authors · 2025-11-19
> **Response to Reviewer mz4J (Part 2)**
>
> > **Weakness 3**: For the results in Table 2 (especially for modification tasks), there is an inconsistency in input modalities. Competing SoTA models (e.g., GPT-4o) receive a natural language prompt plus image or XML input, whereas SimuAgent operates on a Python-dictionary representation. For a fair comparison, all models should be evaluated under identical input formats, or the differences should be clearly justified and analyzed.
>
> We understand the concern about identical input formats. Our intention was to compare the standard industrial workflow of generic models with the workflow that SimuAgent is designed around.
>
> The lightweight Python-dictionary representation is itself a central contribution of our framework, designed to address the token inefficiency and reasoning limitations of raw XML. We therefore view Table 2 as a comparison between:
>
> * **Standard workflow**: A general-purpose model such as GPT-4o operates directly on native engineering artifacts (for example, XML files or screenshots of Simulink models) alongside natural language instructions.
> * **Proposed workflow**: SimuAgent operates on a task-specific, optimized intermediate representation (the Python dictionary), which exposes structure and parameters in a compact format that is easier for an LLM to reason over.
>
> This Python dictionary is generated automatically from the same Simulink models used to produce the XML, discarding only layout and styling information. It does not introduce additional domain knowledge; it restructures the same functional content into a more context-efficient, LLM-friendly form that we regard as part of SimuAgent’s method rather than a neutral preprocessing choice. Evaluating SimuAgent directly on XML would effectively remove this core design element, while feeding GPT-4o our custom dictionary representation would not reflect its typical or recommended usage pattern in industrial settings.
>
> To avoid confusion, we will revise the caption of Table 2 and the description of the experimental setup to:
>
> 1. Clearly state that each model is evaluated under the input modality that best matches its intended workflow.
> 2. Explicitly justify why the Python-dictionary representation is treated as part of SimuAgent’s method rather than an auxiliary preprocessing step.
> 3. Add a short discussion on how these modality choices relate to fairness and what trade-offs they imply, including the fact that our dictionary is an automatically derived, structure-preserving transformation of the same underlying Simulink models.
>
> This should make the comparison more transparent and address the concern about inconsistent input formats.

---

### Official Review · Reviewer_Ehok · 2025-11-04

**Soundness:** 3
**Presentation:** 3
**Contribution:** 3
**Rating:** 8
**Confidence:** 5

**Summary:**

The paper presents SimuAgent for LLM based agent for modeling in Simulink. Key contributions are (1) using the python dictionary structure rather than XML or other token heavy schemes, (2) a two-stage curriculum with a a selfsupervised Abstract–Reconstruct loop and (3) an algorithm ReGRPO that deals with the sparse reward nature of the long horizon problem using self-generated textual reflection traces. It also intruces SimuBench, a large-scale dataset of tasks and show that a Qwen2.5-7B model trained with their pipeline outperforms other baselines (GRPO and GPT-4o). The paper also shows ablation studies to show effect of the two-stage training, reflection and VAE style abstract-reconstruct augmentation.

**Strengths:**

- The paper effectively frames the problem, shows how previous methods (XML) lead to a large number of tokens, and showcases Python-dictionary representation as a suitable choice
- Reflection and retry is a simple mechanism to tackle the sparse reward issue of just having the output of 0/1 at the end of the episode.
- The SimuBench dataset provides examples over various system-design domains.
- The paper is well written, has done extensive experiments, with multiple ablations and transfer to other similar platforms, solidifying the contribution. The figures and plots add to understanding.

**Weaknesses:**

- The algorithm is only compared with GRPO. How does the method compare to other baselines for LLM tool-use and RL?
- Improvements on generic NLP benchmarks are small, code-based tasks show more gain, but SimuBench is the setting where reflection is most helpful.
- More methodological clarifications on reward structure, prompt differences for image-based inputs are needed.

**Questions:**

1. How are the different terms in the reward structure weighted?
2. What minimal hardware is needed for inference/deployment? The manuscript only describes training GPUs and claims laptop-grade GPUs.
3. How can we compare the multi-modal inputs/prompts used for the GPT-4o with the other models?
4. Why does setting the algorithm to Always reflect hurt performance?

---

> ### Author Response · Authors · 2025-11-19
> **Response to Reviewer Ehok (Part 1)**
>
> We express our sincere gratitude to the reviewer for the positive assessment and for recognizing the novelty of our SimuAgent framework, the efficiency of the Python-dictionary representation, and the effectiveness of the ReGRPO algorithm in sparse-reward settings. We are also encouraged that you found the SimuBench dataset to be a valuable contribution to the community.
>
> Below, we address your specific questions and the weaknesses you raised, and provide the requested clarifications.
>
> > **Question 1**: How are the different terms in the reward structure weighted?
>
> We employ a hierarchical reward structure where functional correctness is the dominant factor. All component rewards are first min–max normalized to the range ([0, 1]), and the total reward $R$ is calculated as a weighted sum:
>
> $$R = w_1 \cdot r_{\text{functional}} + w_2 \cdot r_{\text{structural}} + w_3 \cdot r_{\text{executability}} + w_4 \cdot r_{\text{format}}$$
>
> In our experiments, we use a single set of fixed weights that prioritize the final validity of the model:
>
> * $w_1 = 0.5$ (Functional Correctness): Measures simulation accuracy and output values against ground truth (for example, signal similarity).
> * $w_2 = 0.3$ (Structural Similarity): Measures graph similarity (block existence and connection validity) to the reference model.
> * $w_3 = 0.1$ (Executability): Binary reward awarded if the generated Python dictionary compiles and runs in the test harness without errors.
> * $w_4 = 0.1$ (Format / Tool Usage): Adherence to valid JSON syntax and correct tool invocation protocols.
>
> For question-answering tasks, we only use the functional-correctness and format/tool-usage components, as described in Appendix A.3. We will include this specific breakdown in Appendix A.3 of the final revision to ensure full reproducibility.
>
> > **Question 2**: What minimal hardware is needed for inference/deployment? The manuscript only describes training GPUs and claims laptop-grade GPUs.
>
> Our training experiments used H100 GPUs, but the deployment requirements are much lighter, which is a key advantage of fine-tuning a 7B model (Qwen2.5-7B):
>
> * **FP16 inference**: Requires approximately 14–16 GB of VRAM. This fits comfortably on high-end consumer GPUs such as the NVIDIA RTX 4080 or 4090.
> * **Quantized inference (4-bit)**: Requires approximately 6–8 GB of VRAM. This allows SimuAgent to run on standard “laptop-grade” hardware, such as RTX 4060 / 4070 laptop GPUs, or on unified-memory systems such as Apple M2/M3 Max.
>
> In practice, this means SimuAgent can be deployed on a single workstation or modern laptop without any server-class hardware. We will update the Introduction and Section 1 to clearly specify these modern laptop-class hardware configurations for inference and deployment.
>
> > **Question 3**: How can we compare the multi-modal inputs/prompts used for the GPT-4o with the other models?
>
> We evaluated GPT-4o in two distinct, high-context settings to provide a strong baseline, as reported in Table 2. In both cases, we used the same natural-language task description as for SimuAgent and only varied the input modality:
>
> 1. **Image-based setting**: GPT-4o received high-resolution screenshots of the Simulink models, together with the textual instructions.
> 2. **XML-based setting**: GPT-4o received the raw XML file content of the same models with the same instructions.
>
> Even with these rich multi-modal inputs, SimuAgent (51.89%) outperformed GPT-4o (Image: 48.85%, XML: 50.45%) while using only our concise Python-dictionary representation. This supports our hypothesis that the “visual” nature of Simulink, with deeply nested subsystems, hidden parameter dialogs, and complex signal routing, is difficult for vision-language models to fully capture from static screenshots alone.
>
> In contrast, our structured Python representation explicitly exposes hierarchical structure and hidden parameters in a compact and regular format, which appears more amenable to LLM reasoning than raw pixels or verbose XML. We will add this comparison and discussion to Section 4.1, and will also provide the exact prompt templates in Appendix A.3, to make the baseline setup and differences in input modalities clearer.

---

> ### Author Response · Authors · 2025-11-19
> **Response to Reviewer Ehok (Part 2)**
>
> > **Question 4**: Why does setting the algorithm to Always reflect hurt performance?
>
> As shown in Table 5 (Config D, reflection probability $p = 1.0$), setting the algorithm to always reflect degrades performance by 5.64%. We attribute this to two main factors:
>
> 1. **Inefficiency on simple tasks**
>    Reflection is designed to improve exploration on hard, long-horizon tasks by analyzing failures and proposing better plans. Forcing the model to reflect at every step, including trivial ones, injects unnecessary noise and overhead. On simpler tasks, the model often already knows a valid solution, so mandated reflection can distract it from the straightforward path.
>
> 2. **Mismatch with the deployment paradigm**
>    During training, an “always reflect” regime encourages the model to rely on corrective loops instead of learning to produce good plans in a single pass. At inference time, however, our desired mode is one-shot (or low-latency) generation, not a constant retry-reflect cycle. Overusing reflection in training can therefore reduce one-shot performance, because the model expects to be able to “fix itself later” rather than issuing the best possible answer immediately.
>
> We will clarify this trade-off in Section 4.3, emphasizing that reflection is most beneficial when used adaptively rather than as a mandatory step.
>
> > **Weakness 1**: The algorithm is only compared with GRPO. How does the method compare to other baselines for LLM tool-use and RL?
>
> In Table 2, we compare SimuAgent against several standard non-RL baselines: Chain-of-Thought (CoT), retrieval-augmented generation (RAG), and supervised fine-tuning (SFT). These cover common LLM tool-use setups without reinforcement learning.
>
> For RL-style baselines, we made the following design choices:
>
> * We selected **GRPO** as our primary RL baseline because it has become a popular and efficient choice for reasoning-heavy tasks such as math and code, and it avoids the memory and implementation overhead of an explicit critic network.
> * In our preliminary experiments, **standard PPO** was less sample-efficient and more computationally demanding in this token-heavy, long-horizon domain than GRPO’s group-relative baseline.
> * Our tool-use setting essentially follows a **ReAct-style** pattern (interleaved reasoning and acting with tools). SimuAgent extends this by adding a reflection mechanism and structured feedback signals, which improves performance over this ReAct-like baseline.
>
> We will make these choices and comparisons more explicit in the revised text, including a short methodological note on why GRPO and ReAct-style tool use are the most relevant baselines for our setting.
>
> > **Weakness 2**: Improvements on generic NLP benchmarks are small, code-based tasks show more gain, but SimuBench is the setting where reflection is most helpful.
>
> We agree with this observation and view it as consistent with the design goals of our method. SimuAgent and ReGRPO are explicitly optimized for **verifiable, feedback-rich environments**, such as coding and simulation:
>
> * On **generic NLP benchmarks** (for example, MMLU, BBH), where feedback is less structured and rewards are not tightly coupled to program execution, the improvements are indeed modest.
> * On **code benchmarks** such as HumanEval and MBPP, we see stronger gains (+5.9% and +4.5%, respectively).
> * SimuBench, by construction, provides detailed, verifiable signals (functional correctness, structural similarity, executability), which is exactly where the reflection mechanism is most helpful. We also observe successful **zero-shot transfer** of the trained model to other engineering domains such as Modelica and PSCAD.
>
> We will highlight this task-specific nature of the gains more clearly in the revision, emphasizing that SimuAgent is intended primarily as a tool for structured engineering tasks rather than a general-purpose NLP improvement method.
>
> > **Weakness 3**: More methodological clarifications on reward structure, prompt differences for image-based inputs are needed.
>
> The concerns raised here are directly addressed by our clarifications:
>
> * **Reward structure**: Question 1 (above) now provides the exact weights and definitions for each reward component, and we will add the same detailed formula and hyperparameters to Appendix A.3.
> * **Prompt and input differences**: Question 3 (above) explains how GPT-4o was prompted in both the image-based and XML-based settings, and how these setups differ from the Python-dictionary input used for SimuAgent. We will integrate this description into Section 4.1 and add the exact GPT-4o prompt templates to Appendix A.3 so that the baseline configurations can be reproduced.
>
> In the revised manuscript, we will explicitly refer the reader from the main text to the relevant appendix sections where these methodological details are documented, addressing the request for greater clarity and transparency.

---

### Meta-Review · Area_Chair_Baab · 2026-01-13

**Summary:**

This work mainly targets graph-oriented engineering workflows and proposes SimuAgent. SimuAgent is an LLM-powered Simulink agent that uses a concise Python dictionary representation and a plan–execute architecture with Reflection-GRPO (ReGRPO) to improve learning under sparse rewards. Experimental results show that it outperforms Qwen2.5-7B and GRPO, even outperforms GPT-4o in some scenarios.

**Reviewer Concerns:**

1. The choice of baselines is seriously insufficient. The method is only compared with GRPO, without evaluation against mainstream approaches for LLM tool use, agent reinforcement learning, or reflection-based methods.

2. The experimental evaluation is limited. The results may reflect task-specific engineering rather than generalizable improvements, and in some experiments the input modalities are inconsistent across models, potentially leading to unfair comparisons.

3. The methodological description lacks sufficient detail, and some reviewers think the novelty of this work is limited.

**Reviewer Scores:**

After reading the authors’ response, I think the reviewer is likely to keep their score unchanged, as the rebuttal mainly provides clarifications on the claimed novelty and methodological details, but does not offer additional experimental results or baseline comparisons to address the first two concerns. From my perspective, I also agree that the current experimental section remains relatively weak, which limits the contribution of the work.

---

### Decision · Program_Chairs · 2026-01-26

Reject